# LoopServe: An Adaptive Dual-phase LLM Inference Acceleration System for Multi-Turn Dialogues

## Abstract

Multi-turn dialogues are essential in many real-world applications of large language models (LLMs), such as chatbots and virtual assistants. As conversation histories become longer, existing LLMs face increasing computational and memory challenges, which hinder their ability to provide efficient and responsive interactions. Most current acceleration methods either compress the context or optimize key value caching, but they often rely on fixed or position-based heuristics that do not adapt well to the dynamic and unpredictable patterns found in actual multi-turn conversations. As a result, these models cannot accurately identify and prioritize the most relevant context, leading to degraded response quality. In this paper, we present LoopServe, an adaptive dual-phase inference acceleration framework for LLMs in multi-turn dialogues. LoopServe introduces two main innovations. First, it performs online sparsification during the prefilling phase by dynamically selecting the most important parts of the attention matrix for each new input. Second, it uses progressive KV compression during decoding by adaptively maintaining a relevant and efficient cache based on the most recently generated output tokens. We also propose a new benchmark with eleven multi-turn datasets that reflect realistic question positions and conversational dependencies. Extensive experiments demonstrate that LoopServe consistently achieves superior effectiveness compared to existing baselines and significantly accelerates LLM inference across a wide range of long-context dialogue tasks.

## 1 Introduction

Multi-turn dialogues are at the core of numerous real-world applications, from customer service chatbots to virtual assistants and collaborative agents. These scenarios demand that large language models (LLMs) (Hadi et al., 2023; Deng et al., 2025; Li et al., 2024a; Zhou et al., 2024; van Renen et al., 2024) not only generate coherent responses but also maintain contextual consistency across lengthy, evolving conversations. As the number of dialogue turns increases, so does the computational workload. For instance, processing a multi-turn dialogue comprising 10,000 tokens with Llama-3.1-70B (Grattafiori et al., 2024) can demand trillions of floating-point operations (FLOPs), quickly challenging the limits of real-time inference. Despite the remarkable progress of LLMs such as GPT (Brown et al., 2020; Radford et al., 2018; 2019), Llama (Grattafiori et al., 2024; Touvron et al., 2023), and DeepSeek (DeepSeek-AI, 2024; 2025), their inefficiency in handling multi-turn dialogues remains largely unaddressed.

In a typical multi-turn setting, each new user turn expands the conversation history, requiring the LLM to process ever-growing input sequences. The model's self-attention mechanism, which lies at the heart of the Transformer (Vaswani, 2017), needs to compute pairwise attention scores between every pair of tokens in the input. Specifically, given an input sequence of length $n$, an LLM with $P$ parameters and a hidden dimension $d$, the time complexity of generating $m$ tokens is $\mathcal{O}(m \cdot (n+m) \cdot P + m \cdot (n+m)^2 \cdot d)$. With KV Cache, the complexity of prefilling stage is $\mathcal{O}(n \cdot P + n^2 \cdot d)$ without KV Cache, and the complexity of decoding stage is $\mathcal{O}(m \cdot P + m \cdot (n+m) \cdot d)$. For instance, in a 3-turn conversation with 5000 tokens per turn, the effective context length for the model reaches 15,000 tokens, resulting in quadratic growth in both computational cost and memory usage. This compounding effect of context accumulation makes real-time, cost-efficient inference increasingly

difficult as the conversation progresses. Different from single-turn tasks, the context in multi-turn dialogues accumulates dynamically and queries may appear at the beginning, middle, or end of the input, causing attention patterns to shift unpredictably. The resulting attention matrices not only grow with each turn but also exhibit highly dynamic and input-dependent sparsity, exacerbating the inefficiency of current inference methods.

Recent research proposes accelerating LLM inference by reducing the computational burden of attention weight calculations during both the prefilling and decoding stages. In the prefilling stage, where the attention matrix is computed for all token pairs, methods (Jiang et al., 2024a; Lv et al., 2024; Lai et al., 2025), such as Minference (Jiang et al., 2024a), use fixed pattern to sparsify the attention matrix to reduce quadratic computation. During decoding, KV caches store precomputed Key and Value vectors to reduce redundant computation. Methods like H2O (Zhang et al., 2023), SnapKV (Li et al., 2024b), and AdaKV (Feng et al., 2024) cache tokens selected based on tokens at the end of the query. However, these approaches rely on static or position-based heuristics and cannot adapt to the dynamic, input-dependent patterns of real multi-turn dialogues.

Moreover, current evaluation benchmarks (Li et al., 2024a; 2025a; Hsieh et al., 2024; Kim et al., 2025; An et al., 2023) for LLM acceleration misrepresent real-world dialogue scenarios. Most benchmarks (Bai et al., 2024a; Li et al., 2025a; Dacheng Li* & Zhang, 2023) assume queries are always placed at the end of the input and focus on single-turn tasks, which oversimplifies the problem and favors acceleration methods that exploit positional biases. Also, existing multi-turn datasets (He et al., 2024; Fan et al., 2025; Sirdeshmukh et al., 2025; Kwan et al., 2024) are relatively short, with turn lengths typically ranging from 19 to 200 tokens, as detailed in Section A.4.1. As a result, approaches (Zhang et al., 2023; Li et al., 2024b; Feng et al., 2024) that perform well on these benchmarks often fail to generalize to realistic dialogue scenarios where queries may appear at arbitrary positions and contextual dependencies span multiple turns.

In this paper, we propose LoopServe, an adaptive dual-phase LLM inference acceleration framework specifically designed for multi-turn dialogues. LoopServe features two core innovations: online prefilling sparsification and progressive KV compression. In the prefilling phase, LoopServe dynamically identifies and selects the most critical components of the attention matrix, focusing on the vertical and slash line patterns that contribute most to attention weights. Unlike fixed sparsification methods, LoopServe adapts in real time to maintain both efficiency and high attention fidelity. During decoding, LoopServe applies progressive KV compression by dynamically selecting and compressing relevant input tokens based on the most recently generated outputs. This strategy keeps the KV cache efficient and relevant throughout decoding, significantly reducing computational overhead without compromising output quality. We also introduce a multi-turn long-context benchmark containing 11 datasets. This benchmark captures diverse query positions and multi-turn dependencies, offering a more realistic evaluation framework for dialogue scenarios. The contributions of this paper are summarized as follows:

- We empirically reveal that attention patterns and key token positions in multi-turn dialogues are highly dynamic, limiting static sparsification and KV selection.
- We present LoopServe, a dual-phase LLM acceleration framework with online attention sparsification and progressive KV compression, improving multi-turn inference efficiency.
- We introduce a benchmark of 11 long-context multi-turn datasets with varied query positions and dependencies for realistic evaluation.
- Experiments on 11 multi-turn datasets demonstrate the superior performance of LoopServe.

## 2 PRELIMINARY AND RELATED WORK

### 2.1 LARGE LANGUAGE MODELS

LLMs like GPT (Brown et al., 2020), Llama (Grattafiori et al., 2024), and DeepSeek (DeepSeek-AI, 2024; 2025) excel at context understanding and reasoning, enabled by large-scale training and the Transformer architecture (Vaswani, 2017). Transformers are effective due to Multi-Head Self-Attention (MHSA), which captures both local and global token dependencies. Given an input sequence $X = [x_1, x_2, \cdots, x_n]$ with embeddings $\mathbf{X} \in \mathbb{R}^{n \times d}$, the MHSA computes query vectors $\mathbf{Q}^i \in \mathbb{R}^{n \times d_k}$, key vectors $\mathbf{K}^i \in \mathbb{R}^{n \times d_k}$, and value vectors $\mathbf{V}^i \in \mathbb{R}^{n \times d_v}$ for the $i$-th attention head

as $\mathbf{Q}^i = \mathbf{X}\mathbf{W}_{Q^i}, \mathbf{K}^i = \mathbf{X}\mathbf{W}_{K^i}, \mathbf{V}^i = \mathbf{X}\mathbf{W}_{V^i}$, where $\mathbf{W}_{Q^i}$, $\mathbf{W}_{K^i}$, and $\mathbf{W}_{V^i}$ are learnable matrices. Each $i$-th attention head $\mathbf{Z}^i$ as : $\mathbf{Z}^i = \mathsf{Attention}(\mathbf{Q}^i, \mathbf{K}^i, \mathbf{V}^i) = \mathsf{Softmax}\left(\frac{\mathbf{Q}^i(\mathbf{K}^i)^\top}{\sqrt{d_k}}\right)\mathbf{V}^i$. Next, outputs from $h$ heads are concatenated and projected: $\mathbf{Z} = \mathsf{Concat}(\mathbf{Z}^1, \mathbf{Z}^2, \ldots, \mathbf{Z}^h)\mathbf{W}_O$, where $\mathbf{W}_O$ is a learned projection. For text generation, LLMs use an autoregressive process: given $X = [x_1, \ldots, x_n]$, the model predicts the next token $x_{n+1}$ by modeling $P(x_{n+1}|x_1, x_2, \cdots, x_n) = \mathsf{Softmax}(\mathbf{h}_n\mathbf{W}_{\text{out}} + \mathbf{b}_{\text{out}})$, where $\mathbf{h}_n$ is the state at step $n$. The next token $x_{n+1}$ is sampled from this distribution and appended to the sequence. Generation continues until an end-of-sequence token or a maximum length is reached.

## 2.2 Efficient Long-Context Inference

The performance of LLMs degrades with long input contexts, due to the quadratic complexity of self-attention, which scales as $O(Lhn^2)$ for $L$ layers, $h$ heads, and sequence length $n$ (Li et al., 2024a). It makes long-sequence processing prohibitively expensive.

**Context Compression Methods.** Context compression methods reduce the effective sequence length, transforming lengthy inputs into more manageable representations. Filtering-based approaches such as LLMLingua (Jiang et al., 2023), LLMLingua-v2 (Pan et al., 2024), and CompAct (Yoon et al., 2024) focus on identifying and preserving high-relevance content, allowing models to process only critical information. In contrast, RAG-based (retrieval-augmented generation) methods (Zhao et al., 2024; Jiang et al., 2024b; Wang et al., 2025; Chen et al., 2025; Edge et al., 2024) construct knowledge graphs or extract semantic triples from the input, synthesizing them into condensed forms for LLMs. These strategies substantially decrease computational and memory costs, but may sacrifice fine-grained details, potentially affecting output quality.

Also, to reduce computational burden, KV-based approaches minimize the number of attention weight calculations during both prefilling and decoding, summarized in Table 3 in the Appendix.

**Prefilling-stage Optimization.** Self-attention requires $O(n^2)$ computation for an input of length $n$. Recent methods such as Minference (Jiang et al., 2024a), FlexPrefill (Lai et al., 2025), and CritiPrefill (Lv et al., 2024) use binary masks $\mathbf{M} \in \{0,1\}^{n \times n}$ to zero out less important attention weights: $\min \left|\mathbf{A}_i^j - \hat{\mathbf{A}}_i^j\right|, s.t. , \hat{\mathbf{A}}_i^j = \mathsf{Softmax}\left(\frac{\mathbf{Q}^j(\mathbf{K}^j)^\top}{\sqrt{d_k}} - c(1 - \mathbf{M})\right)$, where $c$ is a large constant that suppresses masked entries. This reduces complexity to $O(\alpha n^2)$ per head, with $\alpha \ll 1$. However, Minference (Jiang et al., 2024a) and CritiPrefill (Lv et al., 2024) rely on fixed attention patterns or block selection, while FlexPrefill (Lai et al., 2025) adjusts sparsity globally with simple heuristics. However, our experiments (Section 3) show that attention patterns are highly input-dependent and dynamic, so these static or coarse methods struggle to adapt in multi-turn scenarios.

**Decoding-stage Optimization.** During autoregressive generation, KV cache methods such as H2O (Zhang et al., 2023), SnapKV (Li et al., 2024b), AdaKV (Feng et al., 2024), Quest (Tang et al., 2024), and others (Ge et al., 2024; Li et al., 2024b) select important tokens to store, reducing redundancy. These approaches assume that critical tokens are near the end of the input, performing well on benchmarks like LongBench (Bai et al., 2024a;b), where questions are always placed last. However, as analyzed in Section 3.2, their effectiveness drops when questions appear elsewhere, underscoring the need for adaptive, context-aware KV selection in real dialogue.

## 3 Motivational Experiments and Insights

To clarify the core challenges in accelerating LLM inference for multi-turn dialogues, we conduct motivational experiments in Section 3.1 and 3.2. They reveal how attention patterns are dynamically sparse and how question position influences acceleration effectiveness. Building on these findings, we introduce the LoopServe system, specifically designed to address these real-world challenges.

### 3.1 Key Point 1: Uncertain Attention Patterns

As investigated previously, attention head matrices are highly sparse (Jiang et al., 2024a). Existing acceleration methods (Xiao et al., 2024; Feng et al., 2024; LI et al., 2025; Li et al., 2024b) often rely on sparsifying attention matrices or selecting important KV tokens based on the assumption that

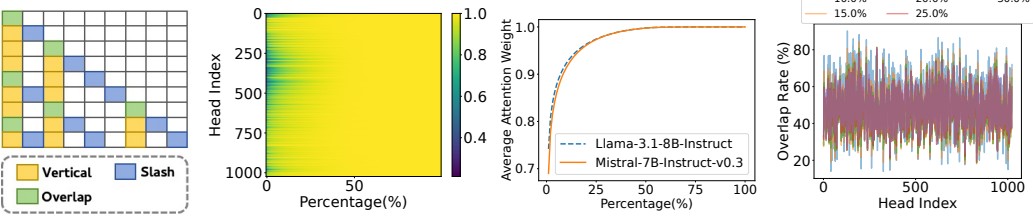

(a) Vertical and slash lines.     (b) All heads of Llama 3.1.     (c) Average ratio on all heads.     (d) Overlap among different inputs.

Figure 1: Attention head sparsity is shown in (b) and (c).

attention patterns are fixed and can be identified offline. In reality, these patterns are highly variable across inputs, heads, and layers, limiting the effectiveness of such approaches, as follows.

**Motivational Observation 1: Only 10% of vertical and slash lines can collectively account for most (e.g., 90%) of the attention weight.** As shown in Figure 1 (a), in an attention matrix, a vertical line (column) indicates a single token attended by all others, common for special tokens like separators or keywords. A slash line (diagonal) shows each token mostly focusing on its nearby tokens, reflecting local attention patterns. We analyze this using the SAMSum QA dataset (Bai et al., 2024a) and the Llama-3.1-8B-Instruct with $n_h$ heads. For each query $X_i$ of length $n_i$, each $k$-th attention matrix $\mathbf{A}_i^k$ contains $n_i$ vertical lines $\mathcal{V}_i^k$ and $n_i$ slash lines $\mathcal{S}_i^k$. The total attention weight for a slash line $s^k$ is $\sum_{(a,b)\in s^k} \mathbf{A}_i^k[a][b]$, and for a vertical line $v^k$, it is $\sum_{(a,b)\in v^k} \mathbf{A}_i^k[a][b]$. We select the top $\eta \cdot 2n_i$ slash and vertical lines $(\hat{\mathcal{S}}_i^k, \hat{\mathcal{V}}_i^k)$ based on their total weights. For each head $k$, we compute the ratio of weight within these lines: $r_i^k = \frac{1}{|\mathcal{D}|}\sum_{X_i\in\mathcal{D}} \frac{\sum_{(a,b)\in\hat{\mathcal{S}}_i^k\cup\hat{\mathcal{V}}_i^k}\mathbf{A}_i^k[a][b]}{n_i}$. Averaging over all $n_h$ heads yields the mean ratio. As shown in Figure 1 (b), higher $\eta$ increases cumulative attention weight. Figure 1 (c) shows that for both Llama and Mistral, just 10% of slash and vertical lines account for 90% of total attention, indicating highly concentrated attention and enabling efficient selection by focusing on these sparse lines.

**Motivational Observation 2: The positions of the top vertical and slash lines within the same head vary across different user inputs.** For a model $M_\theta$ with $n_h$ attention heads and dataset $\mathcal{D} = X_i$, we select the top $\eta \cdot 2n_i$ vertical and slash lines $(\hat{\mathcal{V}}_i^k, \hat{\mathcal{S}}_i^k)$ for each input $X_i$ and attention head $k$. For any pair of queries $X_i$, $X_j$, the overlap of their selected lines under head $k$ is: $r_{i,j}^k = \frac{|\hat{\mathcal{S}}_i^k\cap\hat{\mathcal{S}}_j^k|+|\hat{\mathcal{V}}_i^k\cap\hat{\mathcal{V}}_j^k|}{|\hat{\mathcal{S}}_i^k\cup\hat{\mathcal{S}}_j^k|+|\hat{\mathcal{V}}_i^k\cup\hat{\mathcal{V}}_j^k|}$. Averaging over all heads and input pairs gives the mean overlap ratio: $\frac{1}{n_h|\mathcal{D}|^2}\sum_{k=1}^{n_h}\sum_{X_i,X_j\in\mathcal{D}} r_{i,j}^k$. Using the SAMSum QA dataset (Bai et al., 2024a) and models Llama-3.1-8B-Instruct and Mistral-7B-Instruct-v0.3, with $\eta$ ranging from 0.1 to 0.3, Figure 1 (d) and Figure 6 in Appendix show that for most heads, the overlap remains below 0.5. This indicates that the most important lines differ significantly depending on the input, even within the same head. As a result, important lines cannot be reliably determined offline for use during online inference.

**Motivational Observation 3: For an input $X_i = [C_i^1, C_i^2]$ split into two segments, the top vertical and slash lines within the same head differ between $C_i^1$ and $C_i^2$. Each segment shows its own local attention sparsity pattern.** As illustrated in Figure 2 (a), the key vertical and slash lines in $C_i^1$'s attention matrix are largely absent in $C_i^2$, which displays distinct local patterns. To verify this, we use the SAMSum QA (Bai et al., 2024a) dataset and Llama-3.1-8B-Instruct. For each $X_i$ in dataset $\mathcal{D}$, we split it into $[C_i^1, C_i^2]$ and extract the attention matrices $\mathbf{A}_{C_i^1}^k$ and $\mathbf{A}_{C_i^2}^k$ for each head $k$. After selecting the top-$\eta$ important slash and vertical lines for each $(L_{C_i^1}^k, L_{C_i^2}^k)$ for ( $\mathbf{A}_{C_i^1}^k$, $\mathbf{A}_{C_i^2}^k$), we compute the overlap rate: $r_{C_i^1\to C_i^2}^k = \sum_{l\in L_{C^1}^k} \mathbb{I}(l\in L_{C_i^2}^k)/|L_{C_i^2}^k|$, where $\mathbb{I}$ indicates whether a line from $C_i^1$ is also important in $C_i^2$. Averaging across data gives the mean overlap line rate for each head. Figure 2 (b) shows that, for different $\eta$, the overlap in important lines between $C_i^1$ and $C_i^2$ is consistently low and unstable. This confirms that each segment exhibits unique local attention patterns. This finding indicates that using only a segment (such as the last window or last few tokens) to predict important attention patterns for the whole input is unreliable. As a result, acceleration methods like Minference (Jiang et al., 2024a), SnapKV (Li et al., 2024b), H2O (Zhang et al., 2023), and Keyformer (Adnan et al., 2024), which rely on such assumptions, struggle to deliver consistent performance in real-world scenarios.

### 3.2 KEY POINT 2: QUESTION POSITION MATTERS.

*Motivational experiments indicate that both prefilling based methods and decoding phase acceleration methods, which depend on offline sparse pattern discovery or fixed sparse patterns, tend to*

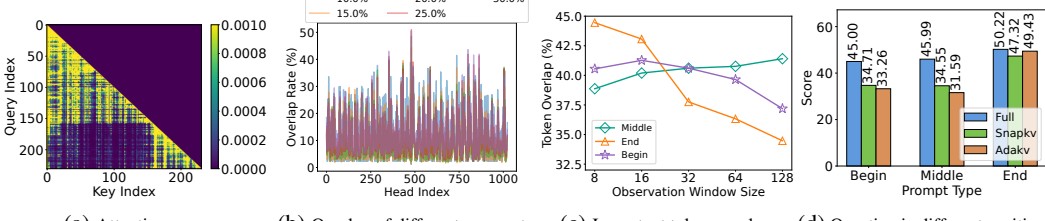

(a) Attention map.    (b) Overlap of different segments.    (c) Important token overlap.    (d) Question in different position.

Figure 2: Different attention sparsity patterns and question position impact on performance.

*underperform in practical scenarios. However, their reported outcomes on existing benchmarks are often similar to those of large language models that use full attention. Why does this discrepancy occur?* The main reason is that benchmarks like Longbench (Bai et al., 2024a;b) always place the user question $q_i$ at the end of the input $X_i = [C_i, q_i]$, so the LLM answers $q_i$ based on context $C_i$. In this setup, acceleration methods only need to focus on the last observation window (near the question), which makes it easier to identify context tokens relevant to the question, thus partially mitigating the unpredictability found in real-world input patterns.

**Motivational Observation 4: Relying only on the last observation window cannot reliably identify important input tokens for generating the output.** Recent methods like H2O (Zhang et al., 2023), SnapKV (Li et al., 2024b), and AdaKV (Feng et al., 2024) select the top-$B$ important tokens $\hat{X}_i$ from $X_i$ based on attention between each input token and the last observation window $X_i^{obs}$ (the last $n_s$ tokens), formally as $\hat{X}_i = \arg\max_{\hat{X}_i \subseteq X_i} \sum_{j=1}^{n_h} \sum_{a \in \hat{X}_i} \sum_{b \in X_i^{obs}} \mathbf{A}_i^k[a][b]$. However, the true top-$B$ tokens $\hat{X}_i^*$ for generating the output $Y_i$ should be selected based on their attention to the output tokens: $\hat{X}_i^* = \arg\max_{\hat{X}_i \subseteq X_i} \sum_{j=1}^{n_h} \sum_{a \in \hat{X}_i} \sum_{b \in Y_i} \mathbf{A}_i^k[a][b]$. We measure the overlap $r_i$ between $\hat{X}_i$ and $\hat{X}_i^*$ ($B$ is set to 10% of $|X_i|$), using Llama-3.1-8B-Instruct and LongEval's topic retrieval set, where the instruction $q_i$ is placed at the beginning, middle, or end of $C_i$. As shown in Figure 2 (c), the average overlap of important tokens is highest when the question is at the end, but much lower when it appears in middle or the beginning. This demonstrates that focusing only on the last part of the input misses relevant information unless the question is placed last.

Similarly, as shown in Figure 2 (d), SnapKV and AdaKV match the original model only when the question is at the end. Their performance drops sharply when the question appears earlier, since they rely on the last tokens for context selection. This shows that current methods are overly dependent on input order and do not generalize well when question positions vary.

## 4 MULTI-TURN LONG-CONTEXT BENCHMARKS

Existing benchmarks, such as NumericBench (Li et al., 2025a), LongBench (Bai et al., 2024a;b), and LongEval (Dacheng Li* & Zhang, 2023) focus on single-turn tasks and place user questions only at the context end, which do not reflect the complexity of real-world, multi-turn conversations (LI et al., 2025). We introduce a benchmark of 11 long-context multi-turn datasets with varied question positions and dependencies. Specifically, each $m$-turn instance is defined as $\mathcal{D} = \mathcal{I}_i = [(C_{i,1}, q_{i,1}, a_{i,1}), \ldots, (C_{i,m}, q_{i,m}, a_{i,m})]|_{i=1}^{|\mathcal{D}|}$, where $C_{i,j}$ is the context at turn $j$ (possibly empty), $q_{i,j}$ is the user question, and $a_{i,j}$ is the LLM-generated answer. We ensure diversity by: (1) **Question Position:** For each turn, the question $q_{i,j}$ can appear at the beginning, end, or between paragraphs of $C_{i,j}$, reflecting more realistic question placements. (2) **Question Relevance:** Answers $a_{i,j}$ may depend on any subset of current and previous contexts $\{C_{i,1}, \ldots, C_{i,j}\}$, with variable subset sizes, simulating diverse real-world dependencies. We construct multi-turn benchmarks for several tasks, including question answering, summarization, and few-shot learning. For construction procedures and detailed benchmark statistics, please refer to Appendix A.4.

## 5 LOOPSERVE SYSTEM

As shown in Figure 3, we propose LoopServe, an adaptive dual-phase system that performs online attention sparsification during the prefilling phase and progressive KV compression during the decoding phase. For an $m$-turn input $\mathcal{I}_i = \{X_{i,j}\}_{j=1}^m$, where $X_{i,j}$ is the context or query in the $j$-th turn, LoopServe generates each answer $y_{i,j}$ using these two steps.

Figure 3: Framework overview of LoopServe.

**Step 1. Online Attention Head Sparsification in Prefilling.** In Algorithm 4 (line 3-5) in Appendix, for each new input $X_{i,j}$ and all inputs $X_i = \left(\cup_{j'=1}^{j-1}(X_{i,j'} \cup y_{i,j'})\right) \cup X_{i,j}$, we get $\hat{X}_{i,j} = [y_{i,j-1}, X_{i,j}]$ as the new appended input in the $j$ turn. Then, for each $k$-th attention head, we first compute the attention matrix $\mathbf{A}_i^k[\hat{X}_{i,j}] \in \mathbb{R}^{\hat{n}_{i,j} \times n_i}$, and then select the slash lines $\hat{\mathcal{S}}_{i,j}^k$ and vertical lines $\hat{\mathcal{V}}_{i,j}^k$ that collectively recover at least $\alpha$ of the total attention weight in $\mathbf{A}_i^k[\hat{X}_{i,j}]$.

**Step 2. Progressive KV Compression in Decoding.** As described in Algorithm 4 (lines 7–14) in Appendix, after every re-selection interval $n_d$ tokens, the framework uses the ProgressiveSelection Algorithm 6 to compute a subset of input tokens $\hat{X}_i^k \subseteq X_i$ for each attention head $k$. these selected tokens $\hat{X}_i^k$ are important for output generation. At each decoding step, LoopServe leverages the compressed KV cache $\{\hat{\mathcal{S}}_{i,j'}^k, \hat{\mathcal{V}}_{i,j'}^k\}_{j'=1,k}^{j,n_h}$ to generate the output sequence $y_{i,j}$.

## 5.1 ONLINE ATTENTION SPARSIFICATION IN PREFILLING

As shown in ***Key Point 1*** in Section 3.1, attention sparsity patterns are highly dynamic, making static or offline selection ineffective. To address this, we propose an online adaptive algorithm that, during prefilling, selects a subset of slash and vertical lines to recover at least an $\alpha$ fraction of the total attention weight for each head, which can be reused in later dialogue turns.

**Definition 1** (Online Prefilling Sparsification Problem). Given the LLM model $M_\theta$ with $n_h$ attention heads, the input $X_i = \left(\bigcup_{j'=1}^{j-1}(X_{i,j'} \cup y_{i,j'})\right) \cup X_{i,j}$, where $y_{i,j'}$ is the answer for $X_{i,j'}$ and $X_{i,j}$ is the current turn's input. We denote the concatenation of the previous answer $y_{i,j-1}$ and the current user input $X_{i,j}$ as $\hat{X}_{i,j} = [y_{i,j-1}, X_{i,j}]$, whose corresponding attention matrix requires sparsification. The $k$-th attention matrix between $X_i$ and $\hat{X}_{i,j}$ is denoted as $\mathbf{A}_i^k[\hat{X}_{i,j}] \in \mathbb{R}^{\hat{n}_{i,j} \times n_i}$. Let $\mathcal{S}_{i,j}^k$ (resp., $\mathcal{V}_{i,j}^k$) denote the set of all slash lines (resp., vertical lines) in $\mathbf{A}_i^k[\hat{X}_{i,j}]$. The goal is to select a subset of slash lines $\hat{\mathcal{S}}_{i,j}^k \subseteq \mathcal{S}_{i,j}^k$ and a subset of vertical lines $\hat{\mathcal{V}}_{i,j}^k \subseteq \mathcal{V}_{i,j}^k$ such that together they recover at least an $\alpha$ fraction of the total attention weight in $\mathbf{A}_i^k[\hat{X}_{i,j}]$, where $\alpha \in [0, 1]$.

$$\min \sum_{s \in \hat{\mathcal{S}}_{i,j}^k} l_s + \sum_{v \in \hat{\mathcal{V}}_{i,j}^k} l_v, \quad s.t. \sum_{(a,b) \in (\hat{\mathcal{S}}_{i,j}^k \cup \hat{\mathcal{V}}_{i,j}^k)} \mathbf{A}_i^k[a][b] \geq \alpha \cdot \hat{n}_{i,j}, \tag{1}$$

where $\hat{n}_{i,j}$ is the total attention weight of the matrix $\mathbf{A}_i^k[\hat{X}_{i,j}]$, and $l_s$ (resp., $l_v$) is the length of the slash line $s$ (resp., vertical line $v$).

**Theorem 1.** *The prefilling sparsification problem is NP-hard. The proof is detailed in Appendix A.5.*

***Algorithm.*** Algorithm 5 in Appendix A.6 takes as input the concatenated sequence $\hat{X}{i,j} = [y_{i,j-1}, X_{i,j}]$, where $y_{i,j-1}$ is the previous answer and $X_{i,j}$ is the current user input. It also requires the $k$-th attention head of the LLM $M_\theta$ and a sparsity threshold parameter $\alpha$. The output consists of the selected slash lines $\hat{\mathcal{S}}_{i,j}^k$ and selected vertical lines $\hat{\mathcal{V}}_{i,j}^k$. The algorithm begins by uniformly sampling a subset $\tilde{X}_{i,j}$ (where $|\tilde{X}_{i,j}| = 48$) from the input $\hat{X}_{i,j}$ to reduce computational cost. It then computes the query matrix $\tilde{\mathbf{Q}}_{i,j}^k$ for $\tilde{X}_{i,j}$ and the key matrix $\mathbf{K}_i^k$ for the full input $X_i$. Next, all slash lines $\mathcal{S}_{i,j}^k$ and vertical lines $\mathcal{V}_{i,j}^k$ are summarized based on $\mathbf{A}_i^k[\tilde{X}_{i,j}]$ and sorted in descending order. Two empty sets, $\hat{\mathcal{S}}_{i,j}^k$ and $\hat{\mathcal{V}}_{i,j}^k$, are initialized to store the selected lines, while the overlap weights $ol_s$ and $ol_v$ are initialized to zero. Algorithm 5 then iteratively selects lines until the total recovered attention weight sum meets or exceeds $\alpha \cdot \text{sum}(\mathbf{A}_i^k[\tilde{X}_{i,j}])$. At each iteration,

it compares the top slash line $s \in \mathcal{S}_{i,j}^k$ and the top vertical line $v \in \mathcal{V}_{i,j}^k$ based on their marginal contributions, $\Delta w_s = w_s - ol_v$ and $\Delta w_v = w_v - ol_s$. Since each slash line overlaps with only one vertical line, $\Delta w_s = w_s - ol_v$. The line with the greater marginal contribution is added to its respective set ($\hat{\mathcal{S}}_{i,j}^k$ or $\hat{\mathcal{V}}_{i,j}^k$), and the overlap weights and total recovered weight are updated accordingly. The loop terminates once the recovery condition is satisfied, and the algorithm returns the sets $\hat{\mathcal{S}}_{i,j}^k$ and $\hat{\mathcal{V}}_{i,j}^k$. The time complexity is $O(n_i|\tilde{X}_{i,j}| + n_i \log n_i + n_i)$, as detailed in Appendix A.6.1.

## 5.2 Progressive KV Compression in Decoding

As shown in Section 3.2, fixed or last-token-focused decoding struggles when questions are not at the end of the input. Thus, we propose a progressive KV compression that selects important input tokens based on recent outputs, leading to greater overlap with truly important input tokens.

To verify this, we consider the LLM $M_\theta$ with $n_h$ attention heads, an input sequence $X_i$, a generated output sequence $Y_i$, and a window size $n_w$. As in Section 3.2, we compute the ground-truth top-$B$ important tokens $\hat{X}_i^* \subseteq X_i$ using the output $Y_i$. We then use observation windows, extracted from either $X_i$ or $Y_i$, to select the top-$B$ important tokens. Specifically, given the window size $n_w$, the $-\frac{|X_i|}{n_w}$-th to $-1$ observation windows from $X_i$ are defined as $X_i[n_i - m \cdot n_w : n_i - (m-1) \cdot n_w]$, $X_i[n_i - (m-1) \cdot n_w : n_i - (m-2) \cdot n_w], ..., X_i[n_i - n_w : n_i]$. The 0-th to $\frac{|Y_i|}{n_w}$ observation windows from $Y_i$ are $Y_i[0 : n_w], Y_i[n_w : 2n_w], ..., Y_i[(m-1) \cdot n_w : m \cdot n_w]$. For each observation window $X_i^{obs}$ from $X_i$ or $Y_i$, we select the top-$B$ tokens $\hat{X}_i \subseteq X_i$ following Section 3.2, and compute the overlap rate between $\hat{X}_i$ and the ground truth $\hat{X}_i^*$ as $\frac{|\hat{X}_i \cap \hat{X}_i^*|}{B}$.

We use LongEval (Dacheng Li* & Zhang, 2023) as in Section 3, experimenting with Llama-3.1-8B-Instruct and setting $B \in \{5\%, 10\%, 15\%\} \cdot |X_i|$. Figure 4(a) shows that important tokens selected from input blocks (block index $< 0$) have low overlap rates, while those selected from output blocks ($\geq 0$) have much higher overlap. This indicates that using output tokens is more effective for identifying relevant input tokens.

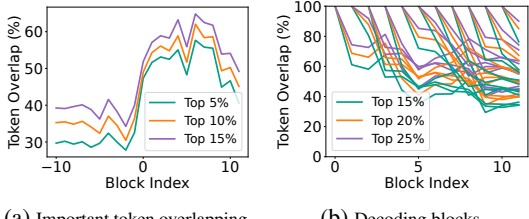

(a) Important token overlapping. (b) Decoding blocks.

Figure 4: Progressive decoding.

We also observe that the overlap of important input tokens between output blocks decreases as the distance between blocks increases, which indicates that we can use the most recent output tokens to select important input tokens for the next output block. Specifically, given the output sequence $Y_i$, we divide it into $n_b$ blocks, i.e., $Y_i = [Y_i^1, \cdots, Y_i^{n_b}]$, where $Y_i^j$ denotes the $j$-th output block of size $\frac{|Y_i|}{n_b}$. For each block $Y_i^j$, we compute the top-$B$ important input tokens as: $\hat{X}_{i,Y_i^j} = \arg\max_{\hat{X}_i \subseteq X_i} \sum_{k=1}^{n_h} \sum_{a \in \hat{X}_i} \sum_{b \in Y_i^j} \mathbf{A}_i^k[a][b]$.

Next, we compare the overlap $\frac{|\hat{X}_{i,Y_i^j} \cap \hat{X}_{i,Y_i^{j'}}|}{B}$ of important input tokens between every pair of output blocks $Y_i^j$ and $Y_i^{j'}$. As shown in Figure 4 (b), for each block $Y_i^j$, the overlap of important input tokens between $Y_i^j$ and $Y_i^{j'}$ (where $j' > j$) gradually decreases as the block distance increases. Notably, the overlap between $Y_i^j$ and its immediate successor $Y_i^{j+1}$ is higher compared to earlier blocks such as $Y_i^{j-2}$. This indicates that tokens identified from $Y_i^j$ are highly relevant for the generation of $Y_i^{j+1}$. Therefore, when generating $Y_i^{j+1}$, we can use the preceding block $Y_i^j$ to dynamically identify the top-$B$ important input tokens. Based on this, we propose the following progressive KV compression algorithm.

**Algorithm.** Algorithm 6 in Appendix firstly set the answer $y_{i,j}$ as empty and decoding step counter $n_o$ to 0 (line 1). During the decoding loop, if the current step reaches either the empirically predefined decoding size 16 or a re-selection interval $n_d$ (line 3), the algorithm extracts the most recent tokens from the input sequence $X_i$, i.e., $X_i^{obs} = X_i[|X_i| - n_d : |X_i|])$, and dynamically updates the compressed KV cache subset $\hat{X}_i^k$ for each $k$-th attention head (line 4-5). Specifically, for each head,

Table 1: Effectiveness Evaluation. The bold number indicates the best performance. Begin, Middle, and End indicate the question positions placed in the user inputs.

| P | | Data | Llama-3.1 | | | | | | | | | Qwen2.5 | | | | | | | | |
|---|---|---|---|---|---|---|---|---|---|---|---|---|---|---|---|---|---|---|---|---|
| | | | Base | H2O | Snap | Ada | SLLM | A-S | T-S | Minf | Ours | Base | H2O | Snap | Ada | SLLM | A-S | T-S | Minf | Ours |
| Begin | QA | MFQA-en | 45.70 | 23.37 | 36.50 | 34.94 | 25.83 | 28.00 | 34.96 | 43.05 | **46.82** | 44.24 | 10.62 | 35.59 | 33.00 | 21.45 | 29.56 | 38.14 | 42.63 | **43.47** |
| | | 2WikiMQA | 31.68 | 20.77 | 29.31 | 28.73 | 22.05 | 19.42 | 24.20 | 28.57 | **35.11** | 37.86 | 11.37 | 34.54 | 33.83 | 23.19 | 22.11 | 32.90 | 35.33 | **37.52** |
| | | Musique | 17.16 | 11.91 | 14.29 | 14.80 | 9.76 | 4.22 | 8.18 | 14.92 | **18.81** | 18.22 | 3.49 | 14.74 | 13.95 | 8.68 | 5.92 | 12.15 | 15.15 | **16.63** |
| | | HotpotQA | 36.67 | 26.49 | 33.87 | 32.98 | 22.52 | 12.72 | 21.29 | 34.38 | **39.00** | 42.93 | 11.63 | 38.90 | 37.56 | 21.42 | 17.22 | 31.03 | 38.67 | **42.50** |
| | | NrtvQA | 13.40 | 6.20 | 13.42 | 8.67 | 5.22 | 7.61 | 9.19 | 13.47 | **14.03** | 13.63 | 4.72 | 11.05 | 10.18 | 6.82 | 7.57 | 10.45 | 10.78 | **14.36** |
| | | Qasper | 25.33 | 10.91 | 20.77 | 17.03 | 16.58 | 20.64 | 23.37 | 22.79 | **24.64** | 26.08 | 7.68 | 21.79 | 20.13 | 17.96 | 19.67 | 23.44 | 24.92 | **25.37** |
| | SUM | MultiNews | 20.08 | 18.18 | 19.20 | 18.03 | 19.30 | 20.10 | 20.15 | **20.25** | 20.14 | 18.39 | 13.50 | 15.87 | 14.52 | 17.29 | 18.29 | 18.30 | 18.30 | **20.14** |
| | | GovReport | 25.25 | 18.58 | 18.36 | 16.53 | 17.98 | 24.16 | 24.13 | 24.55 | **24.90** | 20.93 | 10.76 | 16.80 | 15.86 | 15.90 | 22.28 | 21.90 | 20.86 | **23.60** |
| | | QMSum | 20.53 | 16.53 | 17.49 | 17.56 | 14.66 | 17.77 | 18.92 | 20.15 | **20.65** | 19.97 | 13.44 | 17.54 | 17.39 | 14.66 | 18.45 | 19.12 | 19.61 | **19.66** |
| | FS | TREC | 46.99 | 45.48 | 46.74 | 45.23 | 46.23 | 41.21 | 46.99 | 45.48 | **47.99** | 65.08 | 49.63 | 64.07 | 63.32 | 62.65 | 60.30 | 56.28 | 64.07 | **65.33** |
| | | SAMSUM | 17.32 | 17.80 | 16.75 | 16.48 | 12.95 | 17.39 | 17.35 | 17.46 | **18.12** | 16.19 | 9.46 | 13.97 | 13.15 | 11.22 | 15.32 | 15.92 | 15.96 | **17.15** |
| Middle | QA | MFQA-en | 46.73 | 20.95 | 37.30 | 35.19 | 26.87 | 28.26 | 33.39 | 43.84 | **44.73** | 44.00 | 9.71 | 34.21 | 31.68 | 22.73 | 29.06 | 34.31 | 41.42 | **41.95** |
| | | 2WikiMQA | 34.10 | 16.18 | 31.71 | 29.90 | 25.59 | 18.77 | 27.26 | 32.50 | **34.25** | 27.80 | 7.97 | 22.32 | 22.69 | 14.19 | 20.18 | 26.15 | 25.40 | **27.01** |
| | | Musique | 16.30 | 9.72 | 13.68 | 14.05 | 8.59 | 3.17 | 9.51 | 15.39 | **17.48** | 10.05 | 1.57 | 7.37 | 7.15 | 3.32 | 4.92 | 8.38 | 8.76 | **9.13** |
| | | HotpotQA | 40.63 | 26.94 | 37.48 | 36.30 | 27.30 | 12.36 | 25.25 | 36.81 | **41.25** | 29.43 | 7.88 | 25.24 | 24.56 | 12.96 | 22.00 | 26.43 | | **29.05** |
| | | NrtvQA | 15.29 | 5.55 | 13.06 | 10.64 | 7.26 | 7.14 | 10.10 | 13.98 | **15.25** | 14.82 | 5.15 | 11.88 | 11.83 | 9.04 | 7.68 | 11.24 | 11.75 | **15.21** |
| | | Qasper | 30.79 | 13.43 | 26.80 | 22.07 | 21.40 | 24.90 | 28.86 | 28.47 | **31.12** | 28.74 | 8.02 | 23.57 | 21.77 | 20.75 | 24.73 | 27.84 | 28.72 | **29.27** |
| | SUM | MultiNews | 20.59 | 18.02 | 19.78 | 18.12 | 19.91 | 20.49 | 20.52 | **20.79** | 20.66 | 18.37 | 13.01 | 15.54 | 14.28 | 17.53 | 18.09 | 18.26 | 18.19 | **18.44** |
| | | GovReport | 24.08 | 18.44 | 18.39 | 15.92 | 18.09 | 23.50 | 24.01 | 23.74 | **22.88** | 20.54 | 11.12 | 16.24 | 15.25 | 15.99 | 21.00 | **21.55** | 20.70 | 20.68 |
| | | QMSum | 20.51 | 15.23 | 17.90 | 17.84 | 15.08 | 17.62 | 17.93 | **20.41** | 20.04 | 20.04 | 14.00 | 17.79 | 17.29 | 15.06 | 18.57 | 18.67 | 19.46 | **19.81** |
| | FS | TREC | 50.00 | 41.71 | 50.00 | 48.74 | 50.25 | 50.00 | 50.76 | 52.27 | **56.03** | 64.07 | 39.07 | 62.06 | 60.81 | 63.32 | 63.82 | 40.71 | 64.07 | **65.33** |
| | | SAMSUM | 10.62 | 9.88 | 12.03 | 11.83 | 13.34 | 11.10 | 10.92 | 10.62 | **17.43** | 12.38 | 9.15 | 12.55 | 13.28 | 13.83 | **15.65** | 12.95 | 12.93 | 12.40 |
| End | QA | MFQA-en | 50.93 | 31.76 | 47.93 | 48.38 | 32.52 | 28.62 | 51.40 | 49.67 | **51.69** | 48.82 | 22.64 | 47.57 | 47.24 | 29.28 | 27.66 | 47.94 | **49.18** | 48.67 |
| | | 2WikiMQA | 42.43 | 36.51 | 42.34 | 41.96 | 37.20 | 25.19 | 39.54 | 41.75 | **44.05** | 42.70 | 29.70 | **42.25** | 41.24 | 32.89 | 26.50 | 37.54 | 41.34 | 42.24 |
| | | Musique | 29.39 | 21.79 | 28.07 | 29.01 | 20.96 | 7.80 | 26.44 | 23.56 | **31.60** | 24.18 | 7.68 | 23.08 | 22.39 | 11.82 | 8.68 | 17.65 | 24.34 | **24.82** |
| | | HotpotQA | 53.62 | 45.46 | 52.38 | 53.59 | 43.83 | 25.04 | 51.71 | 51.97 | **55.05** | 53.09 | 25.21 | 51.03 | 50.71 | 35.64 | 24.63 | 43.67 | 53.01 | **53.13** |
| | | NrtvQA | 25.76 | 19.93 | 25.50 | 24.58 | 19.28 | 14.19 | 23.90 | 23.87 | **25.87** | 19.07 | 10.64 | 17.90 | 16.75 | 14.08 | 11.41 | 14.39 | 18.37 | **19.86** |
| | | Qasper | 37.97 | 18.78 | 35.95 | 33.98 | 28.01 | 27.37 | **38.67** | 36.50 | 38.27 | 33.55 | 14.17 | 31.82 | 30.21 | 24.28 | 25.15 | 33.72 | **34.18** | 33.18 |
| | SUM | MultiNews | 20.59 | 18.37 | 20.03 | 18.87 | 19.97 | 20.16 | 20.40 | 20.49 | **20.45** | 18.31 | 14.53 | 16.61 | 15.32 | 17.74 | 17.99 | 18.01 | 18.36 | **18.40** |
| | | GovReport | 23.90 | 19.08 | 20.10 | 18.54 | 18.04 | 23.17 | 23.35 | 23.92 | **24.27** | 21.21 | 8.95 | 18.06 | 17.09 | 16.95 | 21.53 | **21.82** | 21.28 | 21.10 |
| | | QMSum | 22.69 | 19.83 | 22.21 | 22.12 | 19.67 | 18.92 | 22.28 | 22.27 | **22.82** | 21.17 | 17.93 | 20.15 | 20.03 | 18.13 | 18.07 | 20.33 | **21.04** | 20.95 |
| | FS | TREC | 59.05 | 56.53 | 58.54 | 58.54 | 58.54 | 52.51 | 59.55 | 59.05 | **60.31** | 68.34 | 58.29 | 66.08 | 66.59 | 67.84 | 63.82 | 67.59 | **68.85** | 68.34 |
| | | SAMSUM | 18.78 | **24.22** | 23.88 | 20.63 | 19.84 | 18.45 | 17.75 | 17.62 | 23.53 | 39.46 | 37.37 | 39.58 | 38.77 | 38.71 | 39.13 | 39.20 | 39.57 | **39.85** |

we select top-$B$ tokens for each head as: $\hat{X}_i^k = \arg\max_{\hat{X}_i^k \subseteq X_i, |\hat{X}_i^k| = B} \sum_{a \in \hat{X}_i} \sum_{b \in X_i^{obs}} \mathbf{A}_i^k[a][b]$, The LLM generates the next token using the compressed KV cache and the updated input, which is appended to both the input and output sequences (line 6-9). This process iterates until the LLM completes generation, returning the final answer $y_{i,j}$.

# 6 EXPERIMENTS

## 6.1 EXPERIMENTAL SETTINGS

**Datasets, Tasks, and Evaluation Metrics.** We design multi-turn long-context benchmarks. Each instance contains multiple rounds with diverse question positions and dependencies. It covers Question Answering, Summarization, and Few-shot Learning. Dataset statistics and each corresponding metric (e.g., F1, Accuracy, and Rouge-L) are in Table 6 in Appendix A.4.4.

**Baselines.** We compare our LoopServe with seven state-of-the-art KV cache algorithms on two representation LLM base models, including Llama-3.1-8B-Instruct (Grattafiori et al., 2024) and Qwen2.5-7B-Instruct (Team, 2024). The KV cache methods include (1) **H2O** (Zhang et al., 2023), which dynamically retains a balance of recent tokens and heavy-hitter tokens based on accumulated attention scores; (2) **SnapKV (Snap)** (Li et al., 2024b), which compresses KV cache by clustering important positions per attention head; (3) **AdaKV (Ada)** (Feng et al., 2024), which adaptively allocates cache budgets across attention heads; (4) **StreamingLLM (SLLM)** (Xiao et al., 2024), which retains attention sink tokens and recent tokens for stable streaming inference; (5) **A-shape (A-S)** (Xiao et al., 2024), which uses a sparse attention pattern retaining initial and recent tokens; (6) **Tri-Shape (T-S)** (LI et al., 2025), which extends A-shape with diagonal stripes for enhanced context coverage; (7) **Minference (Minf)** (Jiang et al., 2024a), which accelerates pre-filling via dynamic sparse attention with three identified patterns. For baseline details, please refer to Appendix A.7.

**Hyperparameter and Hardware Setting.** All codes are executed on a Rocky Linux 8.10 machine with an 8-core Intel® Xeon® Gold 6542Y CPU, an NVIDIA H100 GPU with 80GB of memory, and 256GB of RAM. For baselines, we use their suggested setting. For main experiments in Section 6.2, for our LoopServe, we set $\alpha = 0.955$, and $n_d = 16$ as defaults, and we set the token budget $B = 1024$ following Li et al. (2024b) Feng et al. (2024) for all baselines and LoopServe.

(a) Generation latency.   (b) Llama-3.1-8B-Instruct.   (c) Llama-3.1-8B-Instruct.   (d) MultiFieldQA.

Figure 5: Efficiency in (a), ablation study in (b), and parameter Sensitivity in (c) and (d).

## 6.2 MAIN EXPERIMENTS

**Effectiveness Evaluation.** To evaluate LoopServe and baselines, we conduct experiments on the proposed 11 multi-turn long-context datasets across three tasks: QA, Summarization (SUM), and Few-shot Learning (FS). For each dataset, we compare LoopServe with six state-of-the-art KV cache acceleration baselines and two base LLMs, using F1, Rouge-L, or Accuracy as appropriate. As shown in Table 1, LoopServe achieves the best or comparable results across most datasets and question positions. Notably, LoopServe maintains strong performance regardless of question location, while baselines like SnapKV and AdaKV perform well only when the question is at the end. This highlights their reliance on positional heuristics, which limits generalization. In contrast, LoopServe's adaptive approach consistently yields higher accuracy and quality, even as context length increases. These gains hold for both Llama-3.1 and Qwen2.5, showing LoopServe generalizes well across LLMs.

**Efficiency Evaluation.** Beyond effectiveness, we also assess LoopServe's generation efficiency. As shown in Figure 5 (a), the efficiency of LoopServe is better than base model and is comparable to baselines. LoopServe is significantly better than the baseline model, as shown in Table 1. This is achieved through efficient online sparsification, which selects only the most critical attention components, and adaptive KV compression, which maintains a compact, relevant cache. Together, these mechanisms reduce computation and memory usage, enabling fast and high-quality generation.

**Ablation Study.** We explore LoopServe-D (progressive KV compression only) and LoopServe-P (online prefilling sparsification only) on three datasets (MF, 2WM, Qsp) using Llama and Qwen. As shown in Figure 5 (b) and Figure 7 in Appendix, LoopServe achieves the best performance, indicating both components are essential and complementary. This advantage holds across tasks, question positions, and model architectures. The ablation study reveals that these two components are complementary: while each addresses a different bottleneck in LLM inference, their combination ensures robust adaptation to diverse input patterns and maximizes both efficiency and accuracy.

## 6.3 PARAMETER SENSITIVITY

We analyze all hyperparameters in LoopServe: attention sparsity threshold $\alpha$ in online prefilling, token budget $B$, and decoding interval $n_d$ in progressive KV compression. Due to space limit, the analysis of the parameter $n_d$ is presented in Appendix A.10.2.

**Threshold $\alpha$ in Equation 1.** The parameter $\alpha$ controls how much total attention weight is preserved in prefilling. Higher $\alpha$ keeps more information but increases computation; lower $\alpha$ boosts efficiency but may lose context. We evaluate LoopServe on 2WikiMQA and Qasper, with Llama and Qwen backbones, across questions at the beginning (-B), middle (-M), and end (-E) positions, varying $\alpha \in \{0.980, 0.985, 0.990, 0.995, 1.00\}$. As shown in Figure 5 (c) and Figure 10 (a) in Appendix, LoopServe get the best accuracy and efficient for $\alpha$ between 0.99 and 1.00. Setting $\alpha$ too low hurts quality, while values close to 1.00 reduce efficiency gains. Overall, LoopServe is not overly sensitive to $\alpha$ within this range, allowing users to balance speed and quality.

**Budget $B$.** The token selection budget $B$ in LoopServe's progressive KV compression controls the trade-off between efficiency and output quality. We evaluate this on MultiFieldQA and Qasper with questions at the begin, middle, and end. In Figure 5 (d) and Figure 10 (b) in Appendix, increasing $B$ improves accuracy by preserving more relevant tokens, but gains are limited beyond 1024 tokens while computation and memory costs rise. Smaller budgets (256 or 512) reduce accuracy, especially for questions at the begin or middle, as important tokens may be missed. End-position questions are less affected since key tokens are already cached. Overall, a budget of 1024–2048 tokens have the best balance of performance and efficiency across all question positions.

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

# A APPENDIX

This appendix is organized as follows: Section A.1 summarizes key notations used throughout the paper. Section A.2 provides a complexity comparison of existing KV cache optimization methods. Section A.3 presents supplementary figures for motivational observations. Section A.4 details the construction and characteristics of our multi-turn LongBench benchmark. Section A.5 presents the formal proof of Theorem 1. Section A.6 provides complete algorithmic descriptions of the LoopServe system. Section A.7 introduces experimental baselines with their configurations in Section A.8. Sections A.9 and A.10 present supplementary experimental results, including ablation studies and parameter sensitivity analysis. Section A.11 evaluates LoopServe on extended 9-turn dialogues. Section A.12 demonstrates effectiveness on smaller models. Section A.13 provides additional efficiency evaluations across varying context and generation lengths. Finally, Section A.14 discloses the usage of LLMs in paper writing.

## A.1 IMPORTANT NOTATIONS TABLE

Table 2: Summary of important notations.

| Symbol | Definition |
|---|---|
| $X_i$ | Input sequence of tokens |
| $X_{i,j}$ | The $j$-turn input of $X_i$ |
| $Y_i$ | Output sequence of tokens |
| $y_{i,j}$ | The $j$-turn output of $X_i$ |
| $n_i, n_{i,j}$ | Length of input sequence $X_i$ and $X_{i,j}$ |
| $m_i, m_{i,j}$ | Length of output sequence $Y_i$ and $y_{i,j}$ |
| $M_\theta$ | LLM model |
| $n_h$ | The total number of attention head of $M_\theta$ |
| $\mathbf{Q}_i^k, \mathbf{K}_i^k, \mathbf{V}_i^k$ | Query, Key, and Value matrices |
| $\mathbf{A}_i^k$ | The $k$-th attention head $X_i$ |
| $\mathcal{S}_i^k, \mathcal{V}_i^k$ | Slash and vertical lines of head $\mathbf{A}_i^k$ |
| $\hat{\mathcal{S}}_i^k, \hat{\mathcal{V}}_i^k$ | Selected slash lines and vertical lines |
| $n_d$ | Decoding interval |
| $B$ | Budget for input tokens |
| $\hat{X}_i^k$ | Selected important tokens for attention head $k$ |

Table 2 provides detailed definitions of important notations appearing in this paper.

## A.2 SUMMARY TABLE OF KV-BASED APPROACHES

Table 3 summarizes the time complexity for each KV-based approach.

## A.3 SUPPLEMENTARY FIGURE FOR MOTIVATIONAL OBSERVATION 2

Figure 6 shows that for most heads, the overlap of Mistral-7B-Instruct-v0.3 remains below 0.5. Please refer to the detailed analysis in Section 3.1 motivational experiment 2.

## A.4 LONG-CONTEXT MULTI-TURN LONGBENCH

Recently, various long-context benchmarks, such as NumericBench (Li et al., 2025a), LongBench (Bai et al., 2024a), and LongEval (Dacheng Li* & Zhang, 2023) have been proposed to evaluate LLMs. However, these benchmarks have two main limitations: (1) They assume user questions always appear at the end of the input, which does not reflect real-world scenarios with questions at arbitrary positions. This bias favors KV-compression methods optimized for end-positioned queries, limiting their generalizability. (2) Most benchmarks are single-turn, overlooking the multi-turn dependencies crucial for realistic conversations (LI et al., 2025).

Table 3: LLM acceleration model comparisons, following LI et al. (2025). P and D denote whether the model has optimization in the Prefilling and Decoding phases, respectively. $n$ is the token size of the input, $m$ is the generation token size, and $c$ and $k$ are constants with $c, k \ll n$ and $c, k \ll m$. We omit the parameter size of LLMs, as it does not affect the understanding of time complexity.

| Methods | P | D | KV Size | Prefilling | Decoding |
|---|---|---|---|---|---|
| **Base LLM** | × | × | - | $O(n^2)$ | $O(nm)$ |
| **LLMLingua** (Pan et al., 2024) | ✓ | × | $O(\alpha n)$ | $O(\alpha^2 n^2)$ | $O(\alpha nm)$ |
| **A-shape** (Xiao et al., 2024) | ✓ | × | $O(n)$ | $O(kn)$ | $O(nm)$ |
| **Tri-shape** (LI et al., 2025) | ✓ | × | $O(n)$ | $O(kn)$ | $O(nm)$ |
| **MInference** (Jiang et al., 2024a) | ✓ | × | $O(n)$ | $O(kn)$ | $O(nm)$ |
| **SLLM** (Xiao et al., 2024) | × | ✓ | $O(k)$ | $O(n^2)$ | $O(km)$ |
| **SnapKV** (Li et al., 2024b) | × | ✓ | $O(k)$ | $O(n^2)$ | $O(km)$ |
| **AdaKV** (Feng et al., 2024) | × | ✓ | $O(k)$ | $O(n^2)$ | $O(km)$ |
| **LoopServe** | ✓ | ✓ | $O(k)$ | $O(kn)$ | $O(k(m-c)+nc)$ |

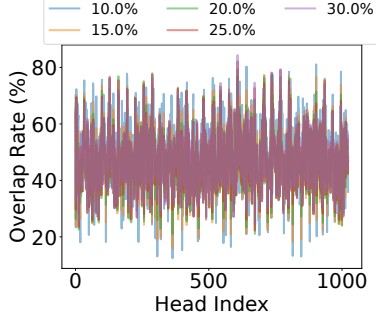

Figure 6: Overlap rate of each head regarding different inputs of Mistral-7B-Instruct-v0.3

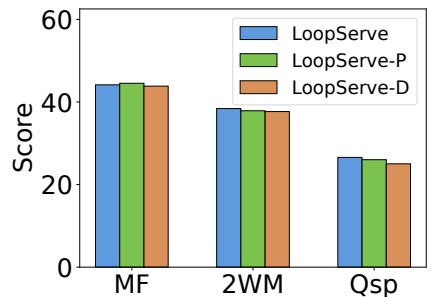

Figure 7: Ablation Study of LoopServe on Qwen2.5-7B-Instruct.

To overcome these issues, we propose a multi-turn benchmark spanning 11 datasets with diverse question positions and interaction patterns, enabling more realistic long-context LLM evaluation.

### A.4.1 COMPARISON WITH EXISTING MULTI-TURN BENCHMARKS

While several multi-turn benchmarks exist, they suffer from overly short user inputs per turn, which causes them to overlook the critical impact of question position (Begin/Middle/End) within long-context inputs on inference acceleration methods in each turn. Table 4 presents a comparison with existing multi-turn datasets.

In contrast, our proposed dataset has three key characteristics:

- **Long-context per turn:** Single-turn inputs exceed 1K tokens, authentically reflecting the computational challenges of long-context reasoning scenarios.

- **Diverse question positions:** Questions are systematically distributed across three positions (Begin/Middle/End) within long-context inputs, enabling evaluation of acceleration methods' adaptability to different question positions.

- **Explicit dependency annotations:** Each question is annotated with which historical contexts it depends on across multiple turns, facilitating analysis of cross-turn context dependencies.

**Empirical Evidence for Non-terminal Question Positions.** To verify the prevalence of non-terminal question positions in real-world applications, we conducted an empirical analysis on the LMSYS-Chat-1M (Zheng et al., 2024) dataset, a real-world multi-turn conversation dataset containing one million conversations between real users and 25 state-of-the-art LLMs, collected from the Vicuna demo and Chatbot Arena website between April and August 2023. We filtered out 334,319

Table 4: Comparison with existing multi-turn benchmarks.

| Dataset | Avg. # Words/Turn | Avg. # Turns |
|---|---|---|
| Multi-IF (He et al., 2024) | 19.02 | 2.99 |
| FairMT-Bench(1K) (Fan et al., 2025) | 29.66 | 5 |
| MT-Bench (Zheng et al., 2023) | 33.49 | 2 |
| MultiChallenge (Sirdeshmukh et al., 2025) | 41.18 | 5.06 |
| MT-Eval (Kwan et al., 2024) | 60.63 | 6.96 |
| StructFlowBench (Li et al., 2025b) | 93.66 | 4.15 |
| Daily-MTD (Tang et al., 2025) | 95.33 | 2.83 |
| MARS-Bench (Yang et al., 2025) | 183.34 | 33.42 |
| **Ours** | **9321.49** | **2.61** |

multi-turn conversations and randomly sampled 10,000 conversations for analysis (totaling 40,118 turns). After excluding single-sentence inputs, we analyzed the position distribution of the remaining 6,676 multi-turn queries.

Table 5: Question position distribution in LMSYS-Chat-1M dataset.

| Question Position | Count | Percentage |
|---|---|---|
| Front | 1,562 | 23.4% |
| Middle | 1,457 | 21.8% |
| End | 3,657 | 54.8% |
| **Non-terminal (Front + Middle)** | **3,019** | **45.2%** |

As shown in Table 5, only 54.8% of questions appear at terminal positions (end: 3,657), while a substantial 45.2% are distributed at non-terminal positions, including front positions (front: 1,562, 23.4%) and middle positions (middle: 1,457, 21.8%). This finding indicates that in real-world long-context multi-turn conversations, nearly half of the questions occur at non-terminal positions in the text, justifying our design of diverse question positions in the benchmark.

### A.4.2 DIVERSE QUESTION POSITIONS

A key distinguishing feature of our benchmark is the systematic incorporation of diverse question positions within long contexts. As illustrated in Figure 8, we design three question placement strategies to reflect real-world scenarios where user questions can appear at arbitrary locations within documents. This design enables comprehensive evaluation of KV cache optimization methods under realistic question distributions, addressing the limitation of existing benchmarks that predominantly assume end-positioned questions.

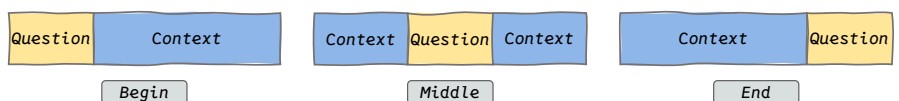

Figure 8: Illustration of diverse question positions in our benchmark. Questions can be positioned at the begin, middle, or end of the context.

### A.4.3 THE DESIGN OF MULTI-TURN LONGBENCH

We represent each $m$-turn long-context data instance in our dataset using a structured format. Specifically, each data instance $\mathcal{I}_i$ consists of $m$ turns, where each turn contains a triplet of context, question, and answer. The complete dataset can be formally denoted as:

$$\mathcal{D} = \{\mathcal{I}_i = [(C_{i,1}, q_{i,1}, a_{i,1}), (C_{i,2}, q_{i,2}, a_{i,2}), \ldots, (C_{i,m}, q_{i,m}, a_{i,m})]\}_{i=1}^{|\mathcal{D}|},$$

where $C_{i,j}$ is the context at the $j$-th turn of the instance $\mathcal{I}_i$, which can be empty, $q_{i,j}$ is the corresponding user question, and $a_{i,j}$ denotes the generated answer of the LLMs for $q_{i,j}$. We design diverse formats for each multi-turn long-context data instance as follows:

- **Diverse Question Positions:** For the $j$-th turn, given a context $C_{i,j} = \{C_{i,j}^1, C_{i,j}^2, \ldots, C_{i,j}^p\}$ consisting of $p$ distinct paragraphs (e.g., segments), the question $q_{i,j}$ can be positioned at various locations within $C_{i,j}$. Specifically, it can appear at the beginning of $C_{i,j}$, at the end of $C_{i,j}$, or between two segments $C_{i,j}^k$ and $C_{i,j}^{k+1}$. Such a way addresses the limitation of existing methods, which only place the question at the end of the context. This placement may not accurately reflect real-world scenarios.

- **Diverse Context Relevance:** At the $j$-th turn, the answer $a_{i,j}$ to question $q_{i,j}$ is derived from the contexts $\{C_{i,j'}\}_{j'=1}^j$. In real user scenarios, the context sources for answering $q_{i,j}$ are diverse. Instead of restricting $q_{i,j}$ to rely solely on $C_{i,j}$, we design the answer $a_{i,j}$ to $q_{i,j}$ to come from a subset of contexts $C_{q_i} \subseteq \{C_{i,j'}\}_{j'=1}^j$, with the size of the subset varying as $|C_{q_i}| \in \{1, 2, \ldots, j\}$.

### A.4.4 MULTI-TURN LONGBENCH GENERATION

Table 6: Multi-turn dataset statistics.

| Type | Dataset | $|D|$ | #Turn | Avg Token | Metric |
|---|---|---|---|---|---|
| | **NQA** | 500 | 3 | 30545.54 | F1 |
| | **Qasper** | 500 | 3 | 5434.41 | F1 |
| **QA** | **MFQA-en** | 500 | 3 | 7279.64 | F1 |
| | **HotpotQA** | 500 | 3 | 13847.19 | F1 |
| | **2WikiMQA** | 500 | 3 | 7471.16 | F1 |
| | **Musique** | 500 | 3 | 16587.56 | F1 |
| | **MultiNews** | 500 | 2 | 2376.46 | Rouge-L |
| **Summary** | **GovReport** | 500 | 2 | 9324.43 | Rouge-L |
| | **QMSum** | 500 | 2 | 12780.29 | Rouge-L |
| **Few-shot** | **TREC** | 199 | 2 | 2293.99 | Accuracy |
| **Learning** | **SAMSUM** | 199 | 2 | 3113.73 | Accuracy |

Based on the above format, we design multi-turn long-context benchmarks across various categories. Dataset details are in Table 6. The construction methodology for QA tasks, summarization tasks, and few-shot learning tasks follows Algorithm 1, Algorithm 2, and Algorithm 3, respectively, as follows:

- **Question Answering (QA).** The algorithm 1 initializes an empty dataset $\mathcal{M}$ (line 1) and iterates $N$ times to construct samples (lines 2-22). In each iteration, it first samples data in lines 3-5: line 3 randomly selects $T$ unique data items from $\mathcal{D}$, while lines 4-5 sample two sets of irrelevant contexts as distractors. Lines 6-21 construct the multi-turn dialogue by initializing an empty list (line 6) and iterating through $T$ turns (lines 7-21). For each turn, lines 8-10 determine dependencies and build context: line 8 randomly samples dependent turn indices, line 9 concatenates contexts from dependent turns, and line 10 appends irrelevant contexts. Lines 11-15 generate question position variants: line 11 extracts the question, line 12 splits context into two parts, and lines 13-15 create three prompt variants (begin, middle, end) by placing the question at different positions. Lines 16-20 store the turn data including prompts, ground truth answer, and dependent turn indices. Line 21 appends the turn data to the multi-turn list, line 22 appends the complete multi-turn dialogue to $\mathcal{M}$, and line 23 returns the final dataset.

- **Summarization.** The algorithm 2 begins by initializing $\mathcal{M}$ (line 1) and iterates $N$ times (lines 2-14). Lines 3-4 sample data: line 3 randomly selects $T$ documents from $\mathcal{D}$, and line 4 samples $T$ irrelevant contexts. Lines 5-7 pre-fragment documents and assign them to turns: the loop iterates through each document (line 5), line 6 splits each document into fragments, and line 7 assigns

these fragments to specific turns, enabling document distribution across the conversation. Lines 8-14 build the multi-turn dialogue by iterating through $T$ turns (line 8). For each turn, lines 9-10 assemble the context: line 9 gathers all fragments assigned up to the current turn, and line 10 appends irrelevant contexts. Line 11 generates a summarization question by concatenating "Summarize: " with the document title. Line 12 creates question position variants (begin, middle, end) using the GenerateQuestionVariants function. Line 34 constructs turn data containing prompts and the ground truth summary. Line 14 appends the turn data to $\mathcal{M}$, and line 15 returns the complete dataset.

- **Few-shot Learning.** The algorithm 3 initializes $\mathcal{M}$ (line 1) and iterates through each sample in $\mathcal{D}$ (lines 2-26). Lines 3-6 extract and validate examples: line 3 extracts few-shot examples from the sample context, and lines 4-5 skip samples with fewer than $2T$ examples to ensure sufficient data. Line 6 partitions examples into $T$ equal groups, assigning each group to a turn. Lines 7-25 build the multi-turn dialogue by initializing an empty list (line 7) and iterating through $T$ turns (lines 8-25). Lines 9-14 generate question position variants: line 9 retrieves examples for the current turn, line 10 splits them at the middle, line 11 generates the task question, and lines 12-14 create three prompt variants by positioning the question differently relative to examples. Lines 15-19 add historical context instructions for turns after the first: line 16 creates an instruction to use previous examples, and lines 17-19 append this instruction to all three prompts. Lines 20-24 store turn data including prompts, ground truth answer, and full dependency indices indicating dependence on all previous turns. Line 25 appends turn data to the multi-turn list, line 26 appends the complete dialogue to $\mathcal{M}$, and line 27 returns the final dataset.

---

**Algorithm 1:** Multi-turn QA Dataset Construction

**Input:** Original single-turn dataset $\mathcal{D}$, turn number $T$, sample number $N$
**Output:** Multi-turn dialogue dataset $\mathcal{M}$

1   $\mathcal{M} \leftarrow \emptyset$
2   **for** $i = 1$ **to** $N$ **do**
     // Step 1:  Sample data
3     data_slice $\leftarrow$ RandomSample($\mathcal{D}$, size $= T$, unique $=$ True)
4     irrelevant_contexts$_1 \leftarrow$ RandomSample($\mathcal{D}$.contexts, size $= T$, exclude $=$ data_slice)
5     irrelevant_contexts$_2 \leftarrow$ RandomSample($\mathcal{D}$.contexts, size $= T$, exclude $=$ data_slice)
     // Step 2:  Construct multi-turn dialogue
6     multi_turn_data $\leftarrow$ [ ]
7     **for** turn_idx $= 0$ **to** $T - 1$ **do**
       // Step 2.1:  Determine dependencies and build context
8       dependent_turns $\leftarrow$ RandomSample($[0, \ldots,$ turn_idx$]$, size $=$ Random($1,$ turn_idx $+ 1$))
9       context $\leftarrow$ ConcatenateContexts(data_slice, dependent_turns)
10      context $\leftarrow$ context $+$ irrelevant_contexts$_1$[turn_idx] $+$ irrelevant_contexts$_2$[turn_idx]
       // Step 2.2:  Generate question position variants
11      question $\leftarrow$ data_slice[turn_idx].question
12      context_parts $\leftarrow$ SplitAtMiddle(context)
13      prompt_begin $\leftarrow$ system_prompt $+$ question $+$ context
14      prompt_middle $\leftarrow$ system_prompt $+$ context_parts[0] $+$ question $+$ context_parts[1]
15      prompt_end $\leftarrow$ system_prompt $+$ context $+$ question
       // Step 2.3:  Store turn data
16      turn_data $\leftarrow$ {
17        prompts : {begin : prompt_begin, middle : prompt_middle, end : prompt_end},
18        ground_truth : data_slice[turn_idx].answer,
19        dependent_turn_indices : dependent_turns
20      }
21      multi_turn_data.append(turn_data)
22     $\mathcal{M}$.append(multi_turn_data)
23   **Return** $\mathcal{M}$

---

**Algorithm 2:** Multi-turn Summarization Dataset Construction

**Input:** Original single-turn dataset $\mathcal{D}$, turn number $T$, sample number $N$
**Output:** Multi-turn dialogue dataset $\mathcal{M}$

1   $\mathcal{M} \leftarrow \emptyset$
2   **for** $i = 1$ **to** $N$ **do**
      // Step 1:  Sample data
3     data_slice $\leftarrow$ RandomSample($\mathcal{D}, T$)
4     irrelevant_contexts $\leftarrow$ RandomSample($\mathcal{D}$.contexts, $T$)
      // Step 2:  Pre-fragment documents and assign to turns
5     **for** doc_idx $= 0$ **to** $T - 1$ **do**
6       fragments[doc_idx] $\leftarrow$ SplitIntoFragments(data_slice[doc_idx].document)
7       assigned_turns[doc_idx] $\leftarrow$ AssignFragmentsToTurns(fragments[doc_idx], doc_idx)
      // Step 3:  Build multi-turn dialogue
8     **for** turn_idx $= 0$ **to** $T - 1$ **do**
9       context $\leftarrow$ AssembleContext(fragments, assigned_turns, turn_idx)
10      context $\leftarrow$ context $+$ irrelevant_contexts[turn_idx]
11      question $\leftarrow$ "Summarize: " $+$ data_slice[turn_idx].document_title
12      prompts $\leftarrow$ GenerateQuestionVariants(question, context)
13      turn_data $\leftarrow$ {prompts, ground_truth : data_slice[turn_idx].summary}
14      $\mathcal{M}$.append(turn_data)

15   **Return** $\mathcal{M}$

---

**Algorithm 3:** Multi-turn Few-shot Learning Dataset Construction

**Input:** Original few-shot dataset $\mathcal{D}$, turn number $T$
**Output:** Multi-turn dialogue dataset $\mathcal{M}$

1   $\mathcal{M} \leftarrow \emptyset$
2   **foreach** sample **in** $\mathcal{D}$ **do**
      // Step 1:  Extract and validate examples
3     examples $\leftarrow$ ExtractExamples(sample.context)
4     **if** Length(examples) $< 2 \times T$ **then**
5       **Continue** // Skip insufficient samples
      // Step 2:  Partition examples across turns
6     examples_per_turn $\leftarrow$ SplitIntoEqualGroups(examples, $T$)
      // Step 3:  Build multi-turn dialogue
7     multi_turn_data $\leftarrow$ []
8     **for** turn_idx $= 0$ **to** $T - 1$ **do**
      // Step 3.1:  Generate question position variants
9       current_examples $\leftarrow$ examples_per_turn[turn_idx]
10      example_parts $\leftarrow$ SplitAtMiddle(current_examples)
11      question $\leftarrow$ GenerateQuestion(sample.input, sample.dataset_type)
12      prompt_begin $\leftarrow$ question $+$ current_examples
13      prompt_middle $\leftarrow$ example_parts[0] $+$ question $+$ example_parts[1]
14      prompt_end $\leftarrow$ current_examples $+$ question
      // Step 3.2:  Add historical context instruction
15      **if** turn_idx $> 0$ **then**
16       history_instruction $\leftarrow$ "Use examples from previous turns with new examples."
17       prompt_begin $\leftarrow$ history_instruction $+$ prompt_begin
18       prompt_middle $\leftarrow$ history_instruction $+$ prompt_middle
19       prompt_end $\leftarrow$ history_instruction $+$ prompt_end
      // Step 3.3:  Store turn data
20      turn_data $\leftarrow$ {
21       prompts : {begin : prompt_begin, middle : prompt_middle, end : prompt_end},
22       ground_truth : sample.answer,
23       dependent_turn_indices : $[0, 1, \ldots, $turn_idx$]$ // Full dependency
24      }
25      multi_turn_data.append(turn_data)
26     $\mathcal{M}$.append(multi_turn_data)

27   **Return** $\mathcal{M}$

---

## A.5  PROOF OF THEOREM 1

*Proof.*  The NP-hardness of the Online Prefilling Sparsification Problem (OPSP) can be established via a reduction from the Partial Set Cover Problem (Chekuri et al., 2019), which is well known to be NP-hard. In the Partial Set Cover Problem, we are given a universe $U = \{u_1, u_2, \ldots, u_m\}$, a collection of subsets $\mathcal{P} = \{P_1, P_2, \ldots, P_n\}$, and a coverage parameter $q \leq |U|$; the goal is to select a subcollection $\mathcal{P}^* \subseteq \mathcal{P}$ of minimum total cost such that the union $\bigcup_{P_i \in \mathcal{P}^*} P_i$ contains at least $q$ elements. To construct the reduction, we map each element $u \in U$ to a specific entry in the attention matrix $\mathbf{A}_i^k[\hat{X}_{i,j}]$ that can be covered in OPSP. Each subset $P_i$ in the Partial Set Cover instance is mapped to a candidate covering line (either a slash line or a vertical line) in OPSP, covering exactly those matrix entries corresponding to the elements of $P_i$. The cost of selecting a subset $P_i$ is mapped to the cost of selecting the corresponding line in OPSP. Covering at least $q$ elements in Partial Set Cover corresponds to the requirement in OPSP to cover at least an $\alpha$ fraction of the total attention weight. Therefore, if we can find a minimum-cost set of lines in OPSP that covers at least an $\alpha$ fraction of the attention weight, we can find a minimum-cost partial cover in the Partial Set Cover problem. As the Partial Set Cover problem is NP-hard, it follows that OPSP is also NP-hard. This reduction demonstrates the equivalence between the partial line covering problem in OPSP and the Partial Set Cover problem, thus justifying the NP-hardness of OPSP.

$\square$

## A.6  ALGORITHMS OF LOOPSERVE SYSTEM

Algorithm 4, Algorithm 5, and Algorithm 6 of LoopServe System in section 5 are listed below.

---

**Algorithm 4:** LoopServe framework overview

---

**Input:**  The $m$-turn input $\mathcal{I}_i = \{X_{i,j}\}_{j=1}^m$, LLM $M_\theta$, threshold $\alpha$, re-selection interval $n_d$, and budget $B$

**Output:**  Answers $\{y_{i,j}\}_{j=1}^m$

1  $X_i \leftarrow \emptyset$
2  **for** $j = 1$ **to** $m$ **do**
3      $X_i = X_i \cup y_{i,j-1} \cup X_{i,j}, \ \hat{X}_{i,j} = [y_{i,j-1}, X_{i,j}]$
    // Step 1:  Parallel Prefilling Line Selection
4      **for** $k = 1$ **to** $n_h$ **do**
5          $\hat{\mathcal{V}}_{i,j}^k, \hat{\mathcal{S}}_{i,j}^k = \mathsf{PrefillingLineSelection}(M_\theta, X_i, \hat{X}_{i,j}, \alpha)$
    // Step 2:  KV Compression for Decoding
6      $\mathcal{L} = \{\hat{\mathcal{V}}_{i,j'}^k, \hat{\mathcal{S}}_{i,j'}^k\}_{j'=1,k=1}^{j,n_h}$
7      $y_{i,j} = \mathsf{ProgressiveDecoding}(M_\theta, X_i, \mathcal{L}, n_d, B)$
8  **Return**  Answers $\{y_{i,j}\}_{j=1}^m$.

---

---

**Algorithm 5:** Adaptive Prefilling Sparsification Framework

**Input:** The input $X_i$ and $\hat{X}_{i,j}$, $k$-th head of LLM $\mathcal{M}_\theta$, the parameter $\alpha$

**Output:** The selected slash lines $\hat{\mathcal{S}}_{i,j}^k$ and vectical lines $\hat{\mathcal{V}}_{i,j}^k$

1   $\tilde{X}_{i,j} = \mathsf{RandomSelect}(\hat{X}_{i,j})$

2   Compute Query $\tilde{\mathbf{Q}}_{i,j}^k$ for $\tilde{X}_{i,j}$

3   Compute Key $\mathbf{K}_i^k$ for $X_i$

4   $\mathbf{A}_i^k[\tilde{X}_{i,j}] = \mathsf{Softmax}\left(\tilde{\mathbf{Q}}_{i,j}^k(\mathbf{K}_i^k)^\top/\sqrt{d_k}\right)$

5   $\mathcal{S}_{i,j}^k = \mathsf{SlashSum}(\mathbf{A}_i^k[\tilde{X}_{i,j}]), \mathcal{V}_{i,j}^k = \mathsf{VerticalSum}(\mathbf{A}_i^k[\tilde{X}_{i,j}])$

6   $\mathcal{S}_{i,j}^k \leftarrow \mathtt{Desc\_Sort}(\mathcal{S}_{i,j}^k), \mathcal{V}_{i,j}^k \leftarrow \mathtt{Desc\_Sort}(\mathcal{V}_{i,j}^k)$

7   $\hat{\mathcal{S}}_{i,j}^k \leftarrow \emptyset, \hat{\mathcal{V}}_{i,j}^k \leftarrow \emptyset$

8   $ol_s = 0, ol_v = 0, sum = 0$

9   **while** $\underline{sum < \alpha \cdot \mathsf{Sum}(\mathbf{A}_i^k[\tilde{X}_{i,j}])}$ **do**

10     $s = \mathcal{S}_{i,j}^k[0], v = \mathcal{V}_{i,j}^k[0]$

11     $\triangle w_s = w_s - ol_v, \triangle w_v = w_v - ol_s$

12     **if** $\underline{\triangle w_s/(l_s - |\hat{\mathcal{V}}_{i,j}^k|) \geq \triangle w_v/(l_v - |\hat{\mathcal{S}}_{i,j}^k|)}$ **then**

13       $\hat{\mathcal{S}}_{i,j}^k = \hat{\mathcal{S}}_{i,j}^k \cup \{s\}, ol_s = ol_s + w_s^{max}$

14       $sum = sum + w_s - ol_v$

15     **else**

16       $\hat{\mathcal{V}}_{i,j}^k = \hat{\mathcal{V}}_{i,j}^k \cup \{v\}, ol_v = ol_v + w_v^{max}$

17       $sum = sum + w_v - ol_s$

18   **Return** $\hat{\mathcal{S}}_{i,j}^k$ and $\hat{\mathcal{V}}_{i,j}^k$.

---

---

**Algorithm 6:** Progressive Decoding

**Input:** Input $\mathcal{I}_i$, LLM $M_\theta$, all selected slash and vertical lines $\{\hat{\mathcal{V}}_{i,j'}^k, \hat{\mathcal{S}}_{i,j'}^k\}_{j'=1,k=1}^{j,n_h}$,
        re-selection interval $n_d$, and budget $B$

**Output:** Answer $y_{i,j}$

1   $n_o = 0, y_{i,j} = \emptyset$

2   **while** LLM generation is not finshed **do**

3     **if** $\underline{n_o = 16 \text{ or } (n_o - 16)\% n_d = 0}$ **then**

4       $X_i^{obs} = X_i[|X_i| - n_d : |X_i|]$

5       **foreach** $\underline{k = 1 \text{ to } n_h}$ **do** $\hat{X}_i^k = \arg\max_{\hat{X}_i^k \subseteq X_i, |\hat{X}_i^k| = B} \sum_{a \in \hat{X}_i} \sum_{b \in X_i^{obs}} \mathbf{A}_i^k[a][b]$ ;

6     $n_o = n_o + 1$

7     $x_{n_i + n_o} = \mathsf{Decoding}(M_\theta, \{\hat{X}_i^k\}_{k=1}^{n_h}, \{\hat{\mathcal{V}}_{i,j'}^k, \hat{\mathcal{S}}_{i,j'}^k\}_{j'=1,k=1}^{j,n_h})$

8     $\hat{X}_i = \hat{X}_i \cup x_{n_i + n_o}, X_i = X_i \cup x_{n_i + n_o}$

9     $y_{i,j} = y_{i,j} \cup x_{n_i + n_o}$

10   **Return** Answer $y_{i,j}$

---

Table 7: Settings used for baseline methods.

| Baseline | Settings |
|---|---|
| LoopServe | $\alpha=0.955, n_d=16, B=1024$ |
| H2O (Zhang et al., 2023) | Recent Size=512, Heavy Hitter Size=512 |
| SnapKV (Li et al., 2024b) | Budget=1024 |
| AdaKV (Feng et al., 2024) | Budget=1024 |
| SLLM (Xiao et al., 2024) | Sliding Window Size=4096 |
| A-Shape (Xiao et al., 2024) | Initial Tokens=128, Local tokens=4096 |
| Tri-Shape (LI et al., 2025) | Initial Tokens=128, Local Tokens=4096, Dense Rows=128 |
| MInference (Jiang et al., 2024a) | We use the best patterns provided by Minference's open code |

### A.6.1 TIME COMPLEXITY ANALYSIS OF ALGORITHM 4

It is $O(n_i|\tilde{X}_{i,j}| + n_i log n_i + n_i)$. Firstly, it takes $O(n_i|\tilde{X}_{i,j}|)$ to compute the partial attention matrix $\mathbf{A}_i^k[\tilde{X}_{i,j}]$. Then, it takes $O(n_i|\tilde{X}_{i,j}|)$ to summarize the values for each slash line and vertical line, and takes $O(n_i \log n_i)$ for descending sorts. Finally, the greedy selection loop runs in $O(n_i)$.

### A.7 EXPERIMENTAL BASELINES INTRODUCTION

We compare our approach against seven state-of-the-art KV cache optimization methods, which employ different strategies including eviction-based policies, compression techniques, and sparse attention patterns to reduce memory consumption during LLM inference.

- **H2O** (Zhang et al., 2023) introduces a KV cache eviction policy based on Heavy-Hitters (H2), which are tokens with the highest accumulated attention scores. It dynamically maintains a balance between recent tokens and H2 tokens by greedily computing attention scores at each decoding step, enabling significant memory reduction while preserving generation quality.

- **SnapKV (Snap)** (Li et al., 2024b) optimizes KV caching through a snapshot mechanism, focusing on reducing redundant computations and enhancing memory utilization by compressing KV cache through clustering important positions per attention head.

- **AdaKV (Ada)** (Feng et al., 2024) dynamically reallocates cache budgets across attention heads based on their attention concentration patterns, shifting resources from heads with concentrated attention to those with dispersed patterns to improve overall cache utilization efficiency.

- **StreamingLLM (SLLM)** (Xiao et al., 2024) maintains both attention sink tokens (initial tokens) and a rolling window of recent tokens in its KV cache to enable stable infinite-length processing without performance degradation in streaming scenarios.

- **A-shape (A-S)** (Xiao et al., 2024) employs a sparse attention pattern that resembles an A shape, retaining attention sink tokens at the beginning and recent tokens at the end to reduce memory footprint while preserving critical contextual information.

- **Tri-Shape (T-S)** (LI et al., 2025) creates a triangular sparse attention pattern for pre-filling by extending A-shape with additional diagonal stripes, enabling enhanced context coverage and more expressive attention computation.

- **Minference (Minf)** (Jiang et al., 2024a) employs dynamic sparse attention by identifying three distinct patterns (A-shape, Vertical-Slash, and Block-Sparse) and building sparse indices online to significantly accelerate the pre-filling stage while maintaining generation quality.

### A.8 BASELINE SETTINGS.

Baseline settings are detailed in Table 7

### A.9 SUPPLEMENTARY RESULTS FOR ABLATION STUDY

Figure 7 shows that LoopServe applied on Qwen2.5-7B-Instruct achieves the best performance, indicating the significance of both components. Please refer to the detailed analysis in ablation study in Section 6.2.

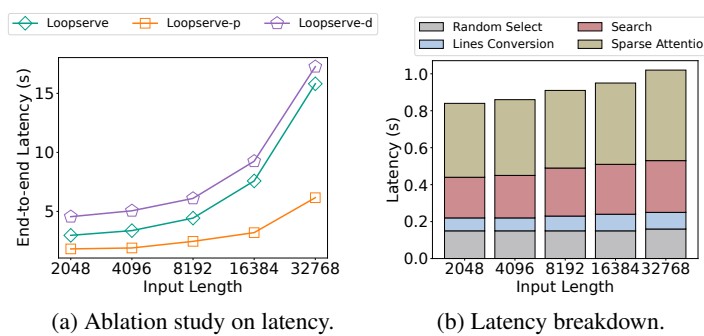

(a) Ablation study on latency.    (b) Latency breakdown.

Figure 9: Ablation study on latency and latency breakdown.

We investigate the inference latency of LoopServe compared to the Base model and two variants: LoopServe-P (sparsification prefilling only) and LoopServe-D (progressive decoding only). As shown in Figure 9 (a), all LoopServe-based methods significantly outperform the Base model. While LoopServe-P yields the lowest latency, the full LoopServe model strikes the optimal balance; it remains faster than LoopServe-D and is substantially more efficient than the baseline, all while ensuring the highest generation quality.

### A.9.1 LATENCY BREAKDOWN.

To assess the latency in different stages for online sparsification prefiling, we conducted microbenchmark on the online sparsification prefilling's attention forward function, and collected total latencies of different stages under different context lengths. We set $\alpha$=99.5%. The results are shown in Figure 9 (b). Sparse Attention remains the dominant factor, accounting for approximately 45-50% of the total inference time; however, its growth remains exceptionally modest. In contrast, Random-Select exhibits almost complete length-invariance, attributed to the low computational overhead of observation window calculation. While the Search component shows a slight latency increase, this overhead is negligible considering the 16x expansion in context window.

### A.10 SUPPLEMENTARY EXPERIMENT FOR PARAMETER SENSITIVITY

### A.10.1 PARAMETER SENSITIVITY TO $\alpha$, BUDGET $B$

Additional results for Parameter sensitivity to $\alpha$ in Figure 10 (a), budget $B$ in Figure 10 (b).

### A.10.2 THE DECODING INTERVAL $n_d$ IN ALGORITHM 6

The decoding interval $n_d$ controls how often LoopServe re-selects important input tokens during progressive KV compression. Smaller $n_d$ values enable frequent adaptation to changing output dependencies, improving accuracy in dynamic dialogues but increasing overhead from more KV cache updates. We evaluate this on MultiNews and GovReport with Llama-3.1-8B-Instruct. and Qwen-2.5-7B-Instruct., covering questions at the beginning, middle, and end. As shown in Figure 10 (c) and (d), moderate $n_d$ values (e.g., 16 or 32) strike the best balance, maintaining efficiency and robust generation quality. Very large $n_d$ reduces adaptivity, leading to lower performance on complex, multi-turn tasks.

### A.10.3 OBSERVATION WINDOW SIZE

To rigorously evaluate the robustness of LoopServe and determine whether the performance gap between methods could be bridged simply by expanding the length of observation window, we conducted a comprehensive study on the observation window size $k$. We compared our method against the state-of-the-art KV-compression baseline, SnapKV, varying $k$ across the set $6, 12, 24, 48, 96$. These experiments were conducted across three datasets featuring diverse user question positions, ensuring a thorough assessment under varying retrieval conditions. All other hyperparameters were maintained consistent with the main experiments. As detailed in Figure 11, the results indicate that

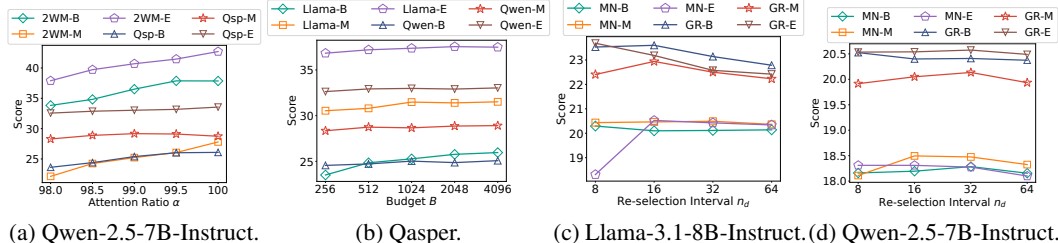

(a) Qwen-2.5-7B-Instruct.  (b) Qasper.  (c) Llama-3.1-8B-Instruct.(d) Qwen-2.5-7B-Instruct.

Figure 10: Parameter sensitivity to $\alpha$ in (a), budget $B$ in (b), and decoding interval $n_d$ in (c) and (d).

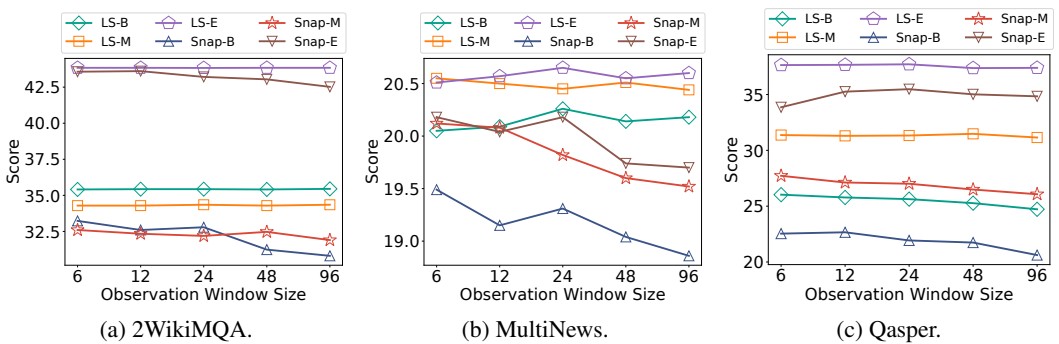

(a) 2WikiMQA.  (b) MultiNews.  (c) Qasper.

Figure 11: Parameter sensitivity to observation window size.

performance is largely insensitive to variations in the observation window size for both LoopServe and SnapKV. Even with a significant increase in $k$ (e.g., from 6 to 96), the performance metrics remain stable. This stability yields a critical insight: simply enlarging the observation window is insufficient to enhance retrieval accuracy or compression quality. Instead, these findings strongly support our hypothesis that the inclusion of output tokens within the observation window—a core design of LoopServe—is the decisive factor for effectively capturing relevant context, rather than merely extending the window size of input tokens.

### A.11  9-TURN QASPER DATASET EVALUATION.

While our primary experiments focused on 3-turn dialogues to balance computational cost with realistic conversation lengths, we further evaluated the robustness of LoopServe on longer interactions. We generated a dataset based on Qasper ranging from 1 to 9 turns and compared LoopServe against strong baselines, including SnapKV, AdaKV, and Minference. All hyperparameter settings remained consistent with the main experiments. As shown in Figure 12, LoopServe consistently outperforms the baselines, demonstrating its effectiveness in capturing inter-turn dependencies even in extended dialogues.

### A.12  EVALUATION ON SMALLER MODEL.

We conduct experiment using Llama-3.2-3B-Instruct, which is more suitable for low-resource devices. We restrict GPU memory to 20GB. We evaluated our method on MultifieldQA, 2WikiMQA and Qasper with question position at the end. The results are shown in Table 8. The results prove that our method is still effective under smaller model sizes, highlighting its abiility to be implemented on various scenarios.

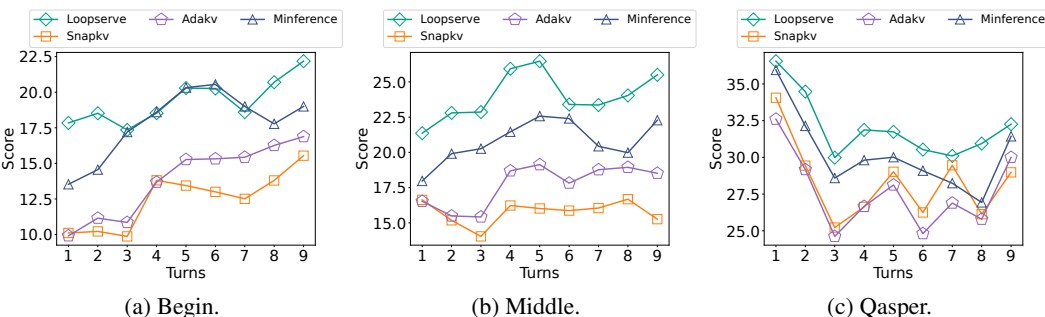

(a) Begin. (b) Middle. (c) Qasper.

Figure 12: Evaluation on 9-turn Qasper dataset.

Table 8: Evaluation on Llama3.2-3B-Instruct.

| Method | MultiFieldQA | 2WikiMQA | Qasper |
|---|---|---|---|
| AdaKV | 45.08 | 34.10 | 25.11 |
| SnapKV | 46.05 | 35.03 | 29.89 |
| Minference | 46.32 | 33.57 | 29.10 |
| LoopServe | **47.16** | **36.21** | **36.30** |

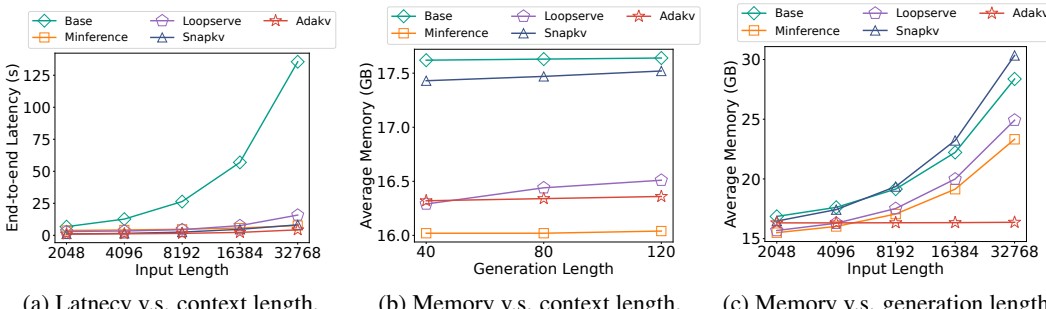

(a) Latnecy v.s. context length. (b) Memory v.s. context length. (c) Memory v.s. generation length.

Figure 13: Efficiency Evaluation.

## A.13 EXTRA EFFICIENCY EVALUATION.

We conducted experiments on end-to-end latency and memory usage under different context length and generation length. We provide a comparison of latency and memory consumption for our LoopServe method and baseline approaches in two settings. We use Llama-3.1-8B-Instruct as backbone LLM and compare three most effective baselines, inclduing SnapKV (Li et al., 2024b), AdaKV (Feng et al., 2024), and Minference (Jiang et al., 2024a). (a) For different input lengths, ranging over {2048, 4096, 8192, 16384, 32768}, the results are shown in Figure 13 (a) and 13 (c). (b) For different decoding lengths, ranging over {40, 80, 120}, the results are shown in 5 (a) and Figure 13 (b). Our LoopServe method is more efficient than the base model and is comparable to AdaKV, SnapKV, and Minference. Nevertheless, in terms of effectiveness, LoopServe significantly outperforms both SnapKV and AdaKV, especially on data where user questions are placed at the beginning or in the middle of user queries. This improvement is due to our progressive KV cache strategy, which maintains important KV pairs rather than selecting only once, as done by AdaKV and SnapKV. The reason for high memory comsumption of SnapKV is that the source code provided by SnapKV doesn't support Grouped Query Attention, causing extra KV Cache storage cost.

## A.14 THE USAGE OF LLMS FOR PAPER WRITING

We use GPT-4o and DeepSeek-R1 to polish our paper.

