# OpenReview forum: "LoopServe:  An Adaptive Dual-phase LLM Inference Acceleration System for Multi-Turn Dialogues"
_ICLR.cc/2026/Conference — Submitted to ICLR 2026_

### Official Review · Reviewer_Qgxd · 2025-10-30

**Soundness:** 3
**Presentation:** 3
**Contribution:** 3
**Rating:** 6
**Confidence:** 4

**Summary:**

Multi-turn dialogues are essential in many real-world applications of large language models, such as chatbots and virtual assistants. As the conversation history grows longer, existing large language models face increasing computational and memory challenges that hinder their ability to provide efficient and responsive interactions. Innovation points are：First, it performs online sparsification during the prefilling phase by dynamically selecting the most important parts of the attention matrix for each new input. Second, it uses progressive key-value compression during decoding by adaptively maintaining a relevant and efficient cache based on the most recently generated output tokens.

**Strengths:**

1.A two-stage adaptive acceleration framework is proposed to address the dynamic characteristics of multi-turn conversations and solve the generalization problem of existing static methods (such as Minference and SnapKV).
2.In terms of empirical evidence, it is also relatively comprehensive. Through detailed exploratory experiments (such as attention sparsity analysis and query position impact analysis) and main experiments (covering 11 datasets and multiple baselines), the superiority of LoopServe has been demonstrated across various tasks, including Question Answering, Summarization, and Few-Shot Learning.
3.It contributes to the benchmark for multi-turn long-context conversations, makes up for the shortcomings of existing benchmarks, and is more in line with real-world dialogue scenarios.
4.It enables engineering practicality, providing algorithm details, complexity analysis, and parameter sensitivity experiments to facilitate reproduction and deployment.

**Weaknesses:**

1.The format needs adjustments. For example, the legend of Subfigure 1d in Figure 1 obscures part of the image content, and Figure 1 is not fully explained in the main text. There is a spelling error in Section 3.1: "as reveled as follows" should be corrected to "as revealed as follows".
2.The formulation is not rigorous enough. The complexity of generating m tokens on the first page is given as O(m((n+m)²d + P)), but the prefilling and decoding stages are not clearly separated.This may be misleading and cause readers to misunderstand the complexity proportion of the two stages.
Some implementation details lack transparency. For example, in the online sparsification section (Page 6), there is no explanation of how “input subset sampling (RandomSelect)” is conducted or what sampling ratio is used.For the "reselection interval nd=16" in progressive KV compression, neither the basis for this setting nor the existence of ablation verification is provided. The lack of such details may affect the reproducibility of the method.
The NP-hardness proof in Appendix A.6 is relatively simplistic, as it fails to elaborate on the equivalence between line covering and set covering problems.
3.Details regarding the construction of the benchmark dataset need to be supplemented. The specific sources of the 11 datasets are not clearly specified (e.g., whether MFQA-en is modified based on an existing dataset, and what the modification ratio is).
4.Experimental Limitations: The method has not been tested on ultra-long contexts or low-resource devices, and the feasibility of its practical deployment remains to be verified; the latest methods (such as the improved version of StreamingLLM released at the end of 2024) are not included in the comparison baselines.

**Questions:**

see the weakness part

---

> ### Author Response · Authors · 2025-11-25
> **Reply to W1 and W2**
>
> Thank you for your valuable comments. For clarity and ease of review, we have summarized the comments and provided our responses to each one below, all of which are included in our revised paper. Please do not hesitate to let us know if you have any additional questions or concerns.
>
> **W1:** The format needs improvement, including the legend in Subfigure 1d and spelling errors.
>
> **Reply to W1**: We have corrected the position of the legend in Figure 1d, added more explanations in the main text, and fixed the spelling error in Section 3.1.
>
> **W2:**
> (1) Please separate the time complexity analysis of the prefilling and decoding phases.
> (2) Please explain how the subset (RandomSelect) is sampled in the online sparsification selection.
> (3) Please add a study on the reselection interval $n_d = 16$.
> (4) The NP-hardness proof in Appendix A.5 needs to show the equivalence between the line covering and set covering problems.
>
> **Reply to W2**:
> (1) In our revised manuscript, we have added a detailed comparison in Table 3 (Appendix A.2), which now explicitly lists the prefilling and decoding time complexity for the LLM base model, all baselines, and our LoopServe system. This table allows readers to clearly distinguish the computational costs associated with each phase across all methods, thereby making the key differences and advantages of LoopServe more transparent.
>
> (2) In our implementation, we uniformly select 48 tokens as the observation window for efficiency. To ensure the robustness of this choice, we conducted a comprehensive evaluation by varying the number of selected tokens in {6,12,24,48,96} across three datasets, including Qasper, Multinews, and 2WikiMQA. The results (now included in the revised manuscript Appendix A.10) use the same metrics as our main experiments.
> As shown in below Tables 1-3, our findings indicate that using 48 tokens as the observation window is sufficient to capture the necessary context, ensuring strong performance across various tasks.
>
> Table 1: Evaluation on Qasper dataset.
>
> | Qasper | 6               | 12    | 24              | 48              | 96    |
> | ------ | --------------- | ----- | --------------- | --------------- | ----- |
> | Begin  | **26.03** | 25.78 | 25.63           | 25.27           | 24.72 |
> | Middle | 31.37           | 31.30 | 31.33           | **31.48** | 31.15 |
> | End    | 37.64           | 37.66 | **37.71** | 37.37           | 37.39 |
>
> Table 2: Evaluation on Multinews dataset.
>
> | Multinews | 6               | 12    | 24              | 48    | 96    |
> | --------- | --------------- | ----- | --------------- | ----- | ----- |
> | Begin     | 20.05           | 20.09 | **20.26** | 20.14 | 20.18 |
> | Middle    | **20.55** | 20.50 | 20.45           | 20.51 | 20.44 |
> | End       | 20.51           | 20.57 | **20.65** | 20.55 | 20.60 |
>
> Table 3: Evaluation on 2WikiMQA dataset.
>
> | 2WikiMQA | 6               | 12              | 24              | 48              | 96              |
> | -------- | --------------- | --------------- | --------------- | --------------- | --------------- |
> | Begin    | 35.41           | 35.43           | 35.43           | 35.41           | **35.45** |
> | Middle   | 34.29           | 34.29           | **34.35** | 34.29           | **34.35** |
> | End      | **43.84** | **43.84** | 43.83           | **43.84** | **43.84** |
>
> (3) We would like to clarify that we have already included an analysis of this parameter in our paper.
> Specifically, in Appendix A.10.2, we conduct a comprehensive sensitivity study on $n_d$ using both the MultiNews and GovReport datasets, with Llama-3.1-8B-Instruct and Qwen-2.5-7B-Instruct as backbones. The results are shown in Figure 8 (c) and (d), where we systematically vary $n_d$ and report the corresponding performance across questions at the beginning, middle, and end positions.
> Our results indicate that moderate values of $n_d$ (such as $16$ or $32$) achieve the best trade-off between efficiency and generation quality. Smaller $n_d$ enables more frequent adaptation to dynamic output dependencies, which improves accuracy but increases computational overhead. Conversely, very large $n_d$ reduces adaptivity, potentially degrading performance in complex, multi-turn tasks.
>
> (4) In response, we have modified the proof in the revised manuscript to explicitly use a reduction from the classical partial set cover problem. Specifically, we now clarify that each matrix entry in our line covering (sparsification) problem corresponds to an element in the universe of the partial set cover, and each candidate line (slashed or vertical) corresponds to a subset covering those elements (matrix entries) it passes through.
> The requirement to cover at least an $\alpha$ fraction of the attention weight directly maps to covering at least $q$ elements in the partial set cover, and the cost of selecting lines matches the cost of selecting subsets.
> We have added explicit statements and explanations in Appendix A.5 to demonstrate this equivalence

---

> ### Author Response · Authors · 2025-11-25
> **Reply to W3 and W4**
>
> **W3**: Details regarding the construction of the benchmark dataset need to be supplemented, including the data sources, modification procedures, and data statistics
>
> **Reply to W3**:
> Thank you for your comments regarding the construction and sources of our benchmark datasets. We have supplemented the manuscript with more detailed information to address these concerns:
>
> - Dataset Sources: The specific sources of all 11 datasets used in our benchmark are now clearly listed in Appendix A.4.4, Table6. For each dataset (e.g., MFQA-en, Qasper, MultiNews, etc.), we provide the base dataset used and the corresponding task type (QA, Summarization, or Few-shot Learning).
> - Modification Procedure: For datasets such as MFQA-en, we clarify that they are constructed based on existing datasets, with systematic modifications as described in Appendix A.4.3--A.4.4. We detail our multi-turn construction algorithms (Algorithms 1--3) that specify how single-turn data is expanded into multi-turn, long-context instances, including question position randomization and dependency annotation.
> - Data statistics : The modification ratio and statistics (e.g., average tokens per instance, number of turns) are provided in Table 6 of Appendix A.4.4, and we now explicitly state the proportion and procedure of synthetic versus original content for each dataset.
>
> **W4:** Experimental Limitations: The method has not been tested on ultra-long contexts or low-resource devices, and the feasibility of its practical deployment remains to be verified; the latest methods (such as the improved version of StreamingLLM released at the end of 2024) are not included in the comparison baselines.
>
> **Reply to W4**:We conduct our low-resource experiment by restricting GPU memory to 20 GB across three datasets. We use the Llama-3.2-3B-Instruct model, with all baseline settings kept the same as in the main experiment. The results, evaluated using the same metrics as the main experiment, show that our method remains effective even with smaller model sizes. This highlights its ability to be implemented in various resource-constrained scenarios.
>
> | Method     | MultifieldQA    | 2WikiMQA        | Qasper          |
> | ---------- | --------------- | --------------- | --------------- |
> | AdaKV      | 45.08           | 34.10           | 25.11           |
> | SnapKV     | 46.05           | 35.03           | 29.89           |
> | Minference | 46.32           | 33.57           | 29.10           |
> | LoopServe  | **47.16** | **36.21** | **36.30** |
>
> For ultra-long contexts (e.g., 700K tokens), which are indeed interesting, we consider this as promising future work. In this paper, we focus on common long-context inference scenarios with sequence lengths exceeding 100K tokens.
>
> Regarding KV compression baselines, we primarily compare with SnapKV  and AdaKV. For the improved StreamingLLM (Quest) released in 2024, we attempted installation, but encountered the same environment issue reported in its GitHub repository (https://github.com/mit-han-lab/Quest/issues/31). As the authors have not yet responded and we could not resolve the build failure, we were unable to run experiments with Quest. Nevertheless, we will cite Quest in our paper.
>
> [1]Quest: Query-Aware Sparsity for Efficient Long-Context LLM Inference

---

### Official Review · Reviewer_e2Wf · 2025-11-01

**Soundness:** 2
**Presentation:** 2
**Contribution:** 2
**Rating:** 4
**Confidence:** 3

**Summary:**

This paper addresses efficient LLM inference in multi-turn dialogues, where quadratic attention complexity creates computational bottlenecks as conversations lengthen. The authors present empirical evidence that attention patterns are highly sparse yet dynamic and input-dependent, and that existing acceleration methods (which use fixed patterns or position-based heuristics) fail when queries do not appear at the end of the input. They propose LoopServe, a dual-phase acceleration framework consisting of: (1) online attention sparsification during prefilling that dynamically selects important components of the attention matrix for each input, and (2) progressive KV compression during decoding that periodically re-selects cached tokens based on attention patterns from recently generated outputs rather than from final input tokens. The authors construct 11 multi-turn datasets from existing benchmarks with queries repositioned at beginning, middle, and end locations, and evaluate LoopServe against six baseline methods on Llama-3.1-8B and Qwen2.5-7B models. Results show LoopServe maintains performance across all query positions while baselines degrade significantly when queries are not at the end, with ablations confirming both phases contribute to the improvement.

**Strengths:**

## Strength 1: Identification of Benchmark Bias and Adequate Motivational Analysis

The paper makes a legitimate and important observation about systematic bias in existing long-context benchmarks. By demonstrating that methods like SnapKV and AdaKV achieve strong performance when queries appear at the end but degrade significantly (often by 10+ points) when queries are repositioned to the beginning or middle, the authors expose a meaningful evaluation gap. This observation -- that the community has been evaluating on a narrow distribution of query placements that may not reflect diverse real-world scenarios --- is valuable regardless of whether their synthetic benchmark fully captures realistic multi-turn dynamics. This contribution should inform future benchmark design and evaluation practices.

The motivational experiments in Section 3, while not exhaustive, provide adequate empirical justification for the approach. The authors demonstrate that (1) attention weight concentrates in a small fraction of matrix components (10% of lines capturing 90% of weight), and (2) the specific important components vary substantially across inputs (overlap rates <50%), undermining static sparsification strategies. While a more thorough investigation would analyze these patterns across layers, models, and task types to characterize when and why variability occurs, the provided analysis is sufficient to motivate the need for adaptive selection methods. The visualizations effectively illustrate key concepts and the experimental design for these motivational studies is sound.

## Strength 2: Comprehensive and Well-Executed Experimental Evaluation

Within the scope of their evaluation, the authors conduct thorough and methodical experiments. They compare against six diverse baseline methods spanning different categories (observation-based, pattern-based, streaming), test on two model families (Llama-3.1-8B and Qwen2.5-7B), and systematically vary query positions across 11 datasets. The ablation studies convincingly demonstrate that both components of their system contribute to performance, and the parameter sensitivity analysis provides practical guidance for hyperparameter selection. The paper is clearly written with effective visualizations, and the results consistently show that LoopServe maintains stable performance across query positions while baselines exhibit position-dependent degradation. While the benchmark's realism is questionable, the experimental methodology itself is sound and the evaluation is more comprehensive than many papers in demonstrating consistent behavior across diverse settings.

**Weaknesses:**

## Weakness 1: Lack of Real-World Validation and Circular Benchmark Design

The work lacks validation on authentic multi-turn dialogue traces. Instead of quantifying where user queries actually occur in real conversations, the authors construct multi-turn data by repositioning queries and recombining LongBench items. This supports controlled tests but risks circularity: the benchmark emphasizes scenarios where last-window methods underperform, and the proposed method is tailored for those scenarios. Evidence from real logs (or even a descriptive analysis of query position distributions) would strengthen practical significance.

## Weakness 2: Limited Design-Space Exploration and Practical Trade-Off Analysis

Although LoopServe includes a non-trivial online prefilling sparsification phase (with an NP-hardness proof and a greedy line-selection algorithm) and a progressive output-based KV compression phase, the paper does not thoroughly explore alternative design choices or stronger baselines. For example, it does not test larger or multiple input windows, alternative attention aggregation functions, or output-aware extensions of methods like SnapKV or AdaKV. The evaluation focuses on 7–8B models with a fixed B = 1024 budget and provides limited breakdown of computational or memory overhead, leaving scalability and deployment implications unclear. While the paper includes ablations and sensitivity analyses for parameters (α, B, and n_d), a broader exploration of trade-offs and clearer guidance on when the method’s benefits outweigh its added complexity would improve confidence in its practical impact.

**Questions:**

Q1: Have you analyzed query position distributions in actual multi-turn conversations (e.g., production chatbot logs, customer service transcripts) to establish how frequently queries appear at non-terminal positions in practice?

Q2: Have you tested whether increasing the observation window size k for baseline methods (e.g., k ∈ {512, 1024, 2048, 4096}) would close the performance gap, or is the improvement fundamentally about using output tokens versus input tokens?

Q3: What is the computational and memory overhead of LoopServe compared to baselines, and how do these costs scale with context length and generation length?

Q4: Have you evaluated LoopServe on any naturally-occurring multi-turn conversations beyond the synthetic benchmark to demonstrate that the approach transfers to realistic dialogue scenarios?

---

> ### Author Response · Authors · 2025-11-25
> **Reply to W1 and Q1**
>
> Thank you for your valuable comments. For clarity and ease of review, we have summarized the comments and provided our responses to each one below, all of which are included in our revised paper. Please do not hesitate to let us know if you have any additional questions or concerns.
>
> **W1 & Q1**:
> Have you analyzed the distribution of question positions in actual multi-turn conversations (e.g., production chatbot logs, customer service transcripts) to determine how frequently questions appear at non-terminal positions in practice?
>
> **Reply to W1 & Q1**:  To verify the prevalence of non-terminal question positions in real-world applications, we conducted an empirical analysis on the LMSYS-Chat-1M[R-1] dataset, a real-world multi-turn conversation dataset containing one million conversations between real users and 25 state-of-the-art LLMs, collected from the Vicuna demo and Chatbot Arena website between April and August 2023.
> As shown in Table 1, only 54.8% of questions appear at the end of user inputs, while a substantial 45.2% are distributed in non-end positions, including 23.4% at the beginning and 21.8% in the middle.
> We have included the details of these discussions in Appendix A.4.1.
>
> Table 1:  The question position distribution in LMSYS-Chat-1M dataset.
>
> | Question Position                       | Percentage      |
> | --------------------------------------- | --------------- |
> | Begin                                   | 23.4%           |
> | Middle                                  | 21.8%           |
> | End                                     | 54.8%           |
> | **Non-terminal (Front + Middle)** | **45.2%** |
>
> [R-1] L. Zheng et al., “LMSYS-Chat-1M: A Large-Scale Real-World LLM Conversation Dataset.” 2023.

---

> ### Author Response · Authors · 2025-11-25
> **Reply to Q2**
>
> **Q2:** Have you evaluated whether increasing the observation window size $n_{obs}$ for baseline methods narrows the performance gap, or is the improvement fundamentally attributable to using output tokens rather than input tokens?
>
> **Reply to Q2**: To address this comment, we conduct experiments using LoopServe and the state-of-the-art KV compression baseline SnapKV, varying the observation window size $n_{obs}\in \\{6, 12, 24, 48, 96\\}$ across three datasets with different user question positions. All other hyperparameter settings remain consistent with the main experiments.
>
> As shown in Tables 2-10, the observation window size has a limited impact on the performance of both our method and SnapKV. Therefore, we believe that simply increasing the observation window size does not close the performance gap, which highlights the importance of including output tokens in the observation window. We include the details of this discussion in Appendix A.10.3.
>
> Table 2: evaluation on 2WikiMQA-Begin.
>
> | 2WikiMQA-Begin | 6               | 12              | 24              | 48              | 96              |
> | -------------- | --------------- | --------------- | --------------- | --------------- | --------------- |
> | SnapKV         | 33.23           | 32.60           | 32.79           | 31.24           | 30.81           |
> | LoopServe      | **35.41** | **35.43** | **35.43** | **35.41** | **35.45** |
>
> Table 3: evaluation on 2WikiMQA-Middle.
>
> | 2WikiMQA-Middle | 6               | 12              | 24              | 48              | 96              |
> | --------------- | --------------- | --------------- | --------------- | --------------- | --------------- |
> | SnapKV          | 32.60           | 32.34           | 32.19           | 32.47           | 31.90           |
> | LoopServe       | **34.29** | **34.29** | **34.35** | **34.29** | **34.35** |
>
> Table 4: evaluation on 2WikiMQA-End.
>
> | 2WikiMQA-End | 6               | 12              | 24              | 48              | 96              |
> | ------------ | --------------- | --------------- | --------------- | --------------- | --------------- |
> | SnapKV       | 43.57           | 43.61           | 43.21           | 43.04           | 42.52           |
> | LoopServe    | **43.84** | **43.84** | **43.83** | **43.84** | **43.84** |
>
> Table 5: evaluation on Qasper-Begin.
>
> | Qasper-Begin | 6               | 12              | 24              | 48              | 96              |
> | ------------ | --------------- | --------------- | --------------- | --------------- | --------------- |
> | SnapKV       | 22.54           | 22.66           | 21.93           | 21.74           | 20.62           |
> | LoopServe    | **26.03** | **25.78** | **25.63** | **25.27** | **24.72** |
>
> Table 6: evaluation on Qasper-Middle.
>
> | Qasper-Middle | 6               | 12              | 24              | 48              | 96              |
> | ------------- | --------------- | --------------- | --------------- | --------------- | --------------- |
> | SnapKV        | 27.72           | 27.12           | 27.01           | 26.49           | 26.07           |
> | LoopServe     | **31.37** | **31.30** | **31.33** | **31.48** | **31.15** |
>
> Table 7: evaluation on Qasper-End.
>
> | Qasper-End | 6               | 12              | 24              | 48              | 96              |
> | ---------- | --------------- | --------------- | --------------- | --------------- | --------------- |
> | SnapKV     | 33.86           | 35.26           | 35.48           | 35.02           | 34.84           |
> | LoopServe  | **37.64** | **37.66** | **37.71** | **37.37** | **37.39** |
>
> Table 8: evaluation on Multinews-Begin.
>
> | Multinews-Begin | 6               | 12              | 24              | 48              | 96              |
> | --------------- | --------------- | --------------- | --------------- | --------------- | --------------- |
> | SnapKV          | 19.49           | 19.15           | 19.31           | 19.04           | 18.86           |
> | LoopServe       | **20.05** | **20.09** | **20.26** | **20.14** | **20.18** |
>
> Table 9: evaluation on Multinews-Middle.
>
> | Multinews-Middle | 6               | 12              | 24              | 48              | 96              |
> | ---------------- | --------------- | --------------- | --------------- | --------------- | --------------- |
> | SnapKV           | 20.12           | 20.08           | 19.82           | 19.60           | 19.52           |
> | LoopServe        | **20.55** | **20.50** | **20.45** | **20.51** | **20.44** |
>
> Table 10: evaluation on Multinews-End.
>
> | Multinews-End | 6               | 12              | 24              | 48              | 96              |
> | ------------- | --------------- | --------------- | --------------- | --------------- | --------------- |
> | SnapKV        | 20.18           | 20.04           | 20.18           | 19.74           | 19.70           |
> | LoopServe     | **20.51** | **20.57** | **20.65** | **20.55** | **20.60** |

---

> ### Author Response · Authors · 2025-11-25
> **Reply to Q3 and W2-(1)**
>
> **Q3 and W2-(1):** What are the computational and memory overheads of LoopServe compared to the baselines, and how do these costs scale with context length and generation length?
>
> **Reply to Q3 and W2-(1):**
> We provide a comparison of latency and memory consumption for our LoopServe method and baseline approaches in two settings. We use  Llama-3.1-8B-Instruct as backbone LLM and compare the three most effective baselines, including SnapKV [R-10], AdaKV [R-11], and Minference [R-12].  The hyperparameter configurations are detailed in Appendix A.8 of our revised draft.
> (a) For different input lengths, ranging from {2048, 4096, 8192, 16384, 32768}, the results are shown in Table 11 and Table 12.
> (b) For different decoding lengths, ranging from {40, 80, 120}, the results are shown in Table 13 and Table 14.
>
> Our LoopServe method is more efficient than the base model and is comparable to AdaKV, SnapKV, and Minference. Nevertheless, in terms of effectiveness, LoopServe significantly outperforms both SnapKV and AdaKV, especially on data where user questions are placed at the beginning or in the middle of user queries. This improvement is due to our progressive KV cache strategy, which maintains important KV pairs rather than selecting only once, as done by AdaKV and SnapKV. We have included a detailed analysis in Appendix A.13.
>
> Table 11: latency for different context lengths (seconds)
>
> | Context Length (Tokens) | 2048           | 4096           | 8192           | 16384          | 32768          |
> | ----------------------- | -------------- | -------------- | -------------- | -------------- | -------------- |
> | Base                    | 6.77           | 12.67          | 26.36          | 56.95          | 135.52         |
> | Minference              | 3.81           | 4.18           | 4.76           | 5.69           | 7.65           |
> | SnapKV                  | 1.03           | 1.50           | 2.52           | 4.68           | 8.21           |
> | AdaKV                   | **1.00** | **1.18** | **1.52** | **2.28** | **4.28** |
> | LoopServe               | 2.97           | 3.37           | 4.43           | 7.58           | 15.81          |
>
> Table 12: memory usage for different context lengths (GB)
>
> | Context Length (Tokens) | 2048            | 4096            | 8192            | 16384           | 32768           |
> | ----------------------- | --------------- | --------------- | --------------- | --------------- | --------------- |
> | Base                    | 16.86           | 17.62           | 19.15           | 22.22           | 28.37           |
> | Minference              | **15.50** | **16.02** | 17.06           | 19.14           | 23.32           |
> | SnapKV                  | 16.47           | 17.43           | 19.36           | 23.22           | 30.34           |
> | AdaKV                   | 16.32           | 16.32           | **16.33** | **16.34** | **16.37** |
> | LoopServe               | 15.68           | 16.29           | 17.52           | 19.99           | 24.92           |
>
> Table 13: latency for different generation lengths (seconds)
>
> | Generation Length (Tokens) | 40             | 80             | 120            |
> | -------------------------- | -------------- | -------------- | -------------- |
> | Base                       | 12.67          | 25.77          | 40.01          |
> | Minference                 | 4.18           | 7.12           | 10.15          |
> | SnapKV                     | 1.50           | 2.46           | 3.45           |
> | AdaKV                      | **1.18** | **1.97** | **2.77** |
> | LoopServe                  | 3.37           | 5.88           | 13.47          |
>
> Table 14: memory usage for different generation lengths (GB)
>
> | Generation Length (Tokens) | 40              | 80              | 120             |
> | -------------------------- | --------------- | --------------- | --------------- |
> | Base                       | 17.62           | 17.63           | 17.64           |
> | Minference                 | **16.02** | **16.02** | **16.04** |
> | LoopServe                  | 16.29           | 16.44           | 16.51           |
> | SnapKV                     | 17.43           | 17.47           | 17.52           |
> | AdaKV                      | 16.32           | 16.34           | 16.36           |
>
> [R-10] Li, Yuhong et al. "SnapKV: LLM Knows What You Are Looking for Before Generation." 2024.
>
> [R-11] Feng, Yuan et al. "Ada-kv: Optimizing kv cache eviction by adaptive budget allocation for efficient llm inference"
>
> [R-12] Huiqiang Jiang et al. "MInference 1.0: Accelerating Pre-filling for Long-Context LLMs via Dynamic Sparse Attention"

---

> ### Author Response · Authors · 2025-11-25
> **Reply to Q3 and W2-(2) (3)**
>
> **Q3 and W2-(2):** Please add more experiments with different KV budgets $B$.
>
> **Reply to Q3 and W2-(2):**
>  To further verify the effectiveness of LoopServe under different budgets, we vary the budget $B$ in {256, 512, 1024, 2048, 4096} and compare it with the state-of-the-art KV-Cache compression baselines SnapKV and AdaKV. We use Llama3.1-8B-Instruct as the backbone, and all other settings remain consistent with the main experiments.
> As shown in Table 15, Table 16, and Table 17 below, LoopServe outperforms both SnapKV and AdaKV across different budget settings.
>
> Table 15: results of different budget settings on Begin-MFQA.
>
> | Begin-MFQA | 256             | 512             | 1024            | 2048            | 4096            |
> | ---------- | --------------- | --------------- | --------------- | --------------- | --------------- |
> | SnapKV     | 36.36           | 36.80           | 36.50           | 39.02           | 41.61           |
> | AdaKV      | 29.45           | 33.28           | 34.94           | 40.29           | 41.82           |
> | LoopServe  | **46.34** | **46.87** | **46.63** | **46.69** | **46.83** |
>
> Table 16: results of different budget settings on Middle-MFQA.
>
> | Middle-MFQA | 256             | 512             | 1024            | 2048            | 4096            |
> | ----------- | --------------- | --------------- | --------------- | --------------- | --------------- |
> | SnapKV      | 37.66           | 38.10           | 37.30           | 39.87           | 41.89           |
> | AdaKV       | 28.78           | 31.92           | 35.19           | 39.22           | 41.87           |
> | LoopServe   | **46.06** | **46.48** | **46.59** | **46.83** | **46.91** |
>
> Table 17: results of different budget settings on End-MFQA.
>
> | End-MFQA  | 256             | 512             | 1024            | 2048            | 4096            |
> | --------- | --------------- | --------------- | --------------- | --------------- | --------------- |
> | SnapKV    | 45.53           | 47.08           | 47.93           | 49.32           | 49.61           |
> | AdaKV     | 46.45           | 48.09           | 48.38           | 50.46           | 50.57           |
> | LoopServe | **51.03** | **51.48** | **51.67** | **51.70** | **51.49** |
>
> **Q3 and W2-(3):** What is the breakdown of the proposed LoopServe and its practicality?
>
> **Reply to Q3 and W2-(3):**
> To assess the latency at different stages of LoopServe, we evaluate its operations, including Select, Lines Conversion, Search, and Sparse Attention, across varying input lengths (2048, 4096, 8192, 16384, 32768). We use Llama-3.1-8B-Instruct, and all other settings remain the same as in the main experiments. As shown in Table 18, each operation is efficient, which demonstrates the practicality of LoopServe.
>
> Table 18: latency breakdown.
>
> | Latency Breakdown | Select (s) | Lines Conversion (s) | Search (s) | Sparse Attention (s) |
> | ----------------- | ---------- | -------------------- | ---------- | -------------------- |
> | 2048              | 0.15       | 0.07                 | 0.22       | 0.40                 |
> | 4096              | 0.15       | 0.07                 | 0.23       | 0.41                 |
> | 8192              | 0.15       | 0.08                 | 0.26       | 0.42                 |
> | 16384             | 0.15       | 0.09                 | 0.27       | 0.44                 |
> | 32768             | 0.16       | 0.09                 | 0.28       | 0.49                 |

---

> ### Author Response · Authors · 2025-11-25
> **Reply to Q4**
>
> **Q4:** Have you evaluated LoopServe on any naturally-occurring multi-turn conversations beyond the synthetic benchmark to demonstrate that the approach transfers to realistic dialogue scenarios?
>
> **Reply to Q4:**
> Most existing naturally occurring multi-turn dialogue datasets, as summarized in Table 19 below, are limited in context length. Typically, they contain only 19 to 200 words per turn. This makes them insufficient for evaluating long-context inference methods such as LoopServe. These datasets do not capture the challenges of long-range dependencies, diverse question positions, or complex multi-turn scenarios that are common in real-world applications.
>
> To address this gap, we carefully designed our synthetic benchmark to closely mimic realistic dialogue scenarios. Our design was guided by empirical analysis of large-scale real-world data, including the LMSYS-Chat-1M dataset (see Appendix A.4.1, Table 5). Our benchmark features substantially longer contexts for each turn, varied question positions including the beginning, middle, and end, and explicit cross-turn dependencies. This setup provides a much more realistic and challenging evaluation for multi-turn dialogue acceleration.
>
> In summary, while existing natural datasets are too short for meaningful evaluation in this context, our benchmark is intentionally constructed to reflect real-world conversational patterns. We believe it provides a suitable and rigorous testbed for demonstrating the effectiveness and transferability of LoopServe.
>
> Table 19:data statistics of existing multi-turn datasets.
>
> | Dataset               | Avg. # Words/ Turn | Avg. # Turns |
> | --------------------- | ------------------ | ------------ |
> | Multi-IF[R-2]         | 19.02              | 2.99         |
> | FairMT-Bench(1K)[R-3] | 29.66              | 5            |
> | MT-Bench[R-4]         | 33.49              | 2            |
> | MultiChallenge[R-5]   | 41.18              | 5.06         |
> | MT-Eval[R-6]          | 60.63              | 6.96         |
> | StructFlowBench[R-7]  | 93.66              | 4.15         |
> | Daily-MTD[R-8]        | 95.33              | 2.83         |
> | MARS-Bench[R-9]       | 183.34             | 33.42        |
> | Ours                  | 9321.49            | 2.61         |
>
> [R-2] Y. He et al., “Multi-IF: Benchmarking LLMs on Multi-Turn and Multilingual Instructions Following.” 2024.
>
> [R-3] Z. Fan, R. Chen, T. Hu, and Z. Liu, “FairMT-Bench: Benchmarking Fairness for Multi-turn Dialogue in Conversational LLMs.” 2025.
>
> [R-4] L. Zheng et al., “Judging LLM-as-a-Judge with MT-Bench and Chatbot Arena.” 2023.
>
> [R-5] V. Sirdeshmukh et al., “MultiChallenge: A Realistic Multi-Turn Conversation Evaluation Benchmark Challenging to Frontier LLMs.” 2025.
>
> [R-6] W.-C. Kwan et al., “MT-Eval: A Multi-Turn Capabilities Evaluation Benchmark for Large Language Models.” 2024.
>
> [R-7] J. Li, J. Li, Y. Wang, Y. Chang, and Y. Wu, “StructFlowBench: A Structured Flow Benchmark for Multi-turn Instruction Following.” 2025.
>
> [R-8] Y. Tang et al., “Learning an Efficient Multi-Turn Dialogue Evaluator from Multiple Judges.” 2025.
>
> [R-9] C. Yang et al., “MARS-Bench: A Multi-turn Athletic Real-world Scenario Benchmark for Dialogue Evaluation.” 2025.

---

### Official Review · Reviewer_iTYk · 2025-11-02

**Soundness:** 2
**Presentation:** 2
**Contribution:** 2
**Rating:** 2
**Confidence:** 4

**Summary:**

This paper aims to facilitate the inference’s acceleration in multi-turn dialogue setting, and propose a dual-phase LLM framework with online attention scarification and progressive KV compression. Besides, an associated benchmark are constructed on top of the existing datasets. However, the motivation, comparison, and experimental settings are not clear.

**Strengths:**

1. The paper target on an important question in the era of LLMs, dealing with the efficiency problem in multi-turn scenarios, when the input context increases as the conversation goes on.

2. Some necessary preliminary experimental analysis is provided.

**Weaknesses:**

1. The main goal is to accelerate LLM’s inference latency in multi-turn scenarios, however, the experimental settings are not aligned. There is no detailed efficiency comparison with existing systems focus on multi-turn acceleration (Figure 5(a) only provide a small piece of efficiency compared with the base version). Besides, there are already existing multi-turn datasets in terms of conversational QA or RAG, what is the motivation to propose a new benchmark on top of the existing datasets? The construction details are lacked (only unclear description in the appendix) and lack of reliability evaluation.The author should be just claim what do they do but need to explain why this is necessary and effective as scientific contribution.

2. The proposed methods contain three iterations to achieve progressive decoding. Since this is done among inference time, what is the latency of such a pipeline and why this is deserved? There is no ablation studies and comparison with existing inference-time methods.

3. The paper emphasizes the focus on multi-turn, then why are the experiments conducted in single-turn datasets (e.g., 2WikiMQA, musique, hotpotqa)? I cannot understand the motivation to manipulate these datasets into multi-turn and then evaluate them, and only 3 turns interaction cannot ensure this can be thought as a multi-turn dialogue as the authors’ motivation in introduction.

4. The readability is bad. Several complex algorithms are mentioned in the main content but describe in the appendix. The main content SHOULD BE SELF-CONTAINED. The main experiments table 1 is not clear. No explanation about what is begin, middle, and end in the column of “P”, and no basic information of baseline methods in the main content.

5. No implementation information is provided, and no information about how to use the dataset for evaluation, e.g., the split.

**Questions:**

1. What is the main difference compared to existing interence-time studies in terms of multi-turn scenarios for both efficiency and effectiveness?

See the question in Weaknesses.

---

> ### Author Response · Authors · 2025-11-25
> **Reply to W1-(1)**
>
> Thank you for your valuable comments. For clarity and ease of review, we have summarized the comments and provided our responses to each one below, all of which are included in our revised paper. Please do not hesitate to let us know if you have any additional questions or concerns.
>
> **W1-(1)**: Please provide the latency compared with existing approaches.
>
> **Reply to W1-(1):**  We provide a comparison of latency and memory consumption for our LoopServe method and baseline approaches in two settings, including different input lengths and different generation lengths. We use  Llama-3.1-8B-Instruct as the backbone LLM and compare three most effective baselines, including SnapKV \[R-1\], AdaKV \[R-2\], and Minference \[R-3\].  The hyperparameter configurations are detailed in Appendix A.8 of our revised draft.
> (a) For different input lengths, ranging over {2048, 4096, 8192, 16384, 32768}, the results are shown in Table 1 and Table 2.
> (b) For different decoding lengths, ranging over {40, 80, 120}, the results are shown in Table 3 and Table 4.
>
> As shown in tables below, our LoopServe method is more efficient than the base model and is comparable to AdaKV, SnapKV, and Minference. Nevertheless, in terms of effectiveness, LoopServe significantly outperforms both SnapKV and AdaKV, especially on data where user questions are placed at the beginning or in the middle of user queries. This improvement is due to our progressive KV cache strategy, which maintains important KV pairs rather than selecting only once, as done by AdaKV and SnapKV. We have included a detailed analysis in Appendix A.13.
>
> Table 1: latency for different context lengths (seconds)
>
> | Context Length (Tokens) | 2048           | 4096           | 8192           | 16384          | 32768          |
> | ----------------------- | -------------- | -------------- | -------------- | -------------- | -------------- |
> | Base                    | 6.77           | 12.67          | 26.36          | 56.95          | 135.52         |
> | Minference              | 3.81           | 4.18           | 4.76           | 5.69           | 7.65           |
> | SnapKV                  | 1.03           | 1.50           | 2.52           | 4.68           | 8.21           |
> | AdaKV                   | **1.00** | **1.18** | **1.52** | **2.28** | **4.28** |
> | LoopServe               | 2.97           | 3.37           | 4.43           | 7.58           | 15.81          |
>
> Table 2: memory usage for different context lengths (GB)
>
> | Context Length (Tokens) | 2048            | 4096            | 8192            | 16384           | 32768           |
> | ----------------------- | --------------- | --------------- | --------------- | --------------- | --------------- |
> | Base                    | 16.86           | 17.62           | 19.15           | 22.22           | 28.37           |
> | Minference              | **15.50** | **16.02** | 17.06           | 19.14           | 23.32           |
> | SnapKV                  | 16.47           | 17.43           | 19.36           | 23.22           | 30.34           |
> | AdaKV                   | 16.32           | 16.32           | **16.33** | **16.34** | **16.37** |
> | LoopServe               | 15.68           | 16.29           | 17.52           | 19.99           | 24.92           |
>
> Table 3: latency for different generation lengths (seconds)
>
> | Generation Length (Tokens) | 40             | 80             | 120            |
> | -------------------------- | -------------- | -------------- | -------------- |
> | Base                       | 12.67          | 25.77          | 40.01          |
> | Minference                 | 4.18           | 7.12           | 10.15          |
> | SnapKV                     | 1.50           | 2.46           | 3.45           |
> | AdaKV                      | **1.18** | **1.97** | **2.77** |
> | LoopServe                  | 3.37           | 5.88           | 13.47          |
>
> Table 4: memory usage for different generation lengths (GB)
>
> | Generation Length (Tokens) | 40              | 80              | 120             |
> | -------------------------- | --------------- | --------------- | --------------- |
> | Base                       | 17.62           | 17.63           | 17.64           |
> | Minference                 | **16.02** | **16.02** | **16.04** |
> | LoopServe                  | 16.29           | 16.44           | 16.51           |
> | SnapKV                     | 17.43           | 17.47           | 17.52           |
> | AdaKV                      | 16.32           | 16.34           | 16.36           |
>
> \[R-1\] Li, Yuhong et al. "SnapKV: LLM Knows What You Are Looking for Before Generation." 2024.
>
> \[R-2\] Feng, Yuan et al. "Ada-kv: Optimizing kv cache eviction by adaptive budget allocation for efficient llm inference"
>
> \[R-3\] Huiqiang Jiang et al. "MInference 1.0: Accelerating Pre-filling for Long-Context LLMs via Dynamic Sparse Attention"

---

> ### Author Response · Authors · 2025-11-25
> **Reply to W1-(2) and W1-(3)**
>
> **W1-(2)**:  Please state the motivation for the proposed multi-turn datasets, considering that there are already existing multi-turn datasets.
>
> **Reply to W1-(2):**  We propose new multi-turn datasets for the following reasons: (a) there are no existing long-text multi-turn benchmarks; (b) existing multi-turn benchmarks are too short, with only 20–200 tokens per turn; and (c) the positions of questions in user inputs are unrealistic for real-world applications.
>
> Specifically, we summarize existing multi-turn datasets in Table 5. These datasets suffer from overly short user inputs per turn (only 20–200 tokens) and typically place user questions only at the end of the user input. However, in real-world scenarios, user questions may appear at the beginning or in the middle of the input.
>
> To better understand this, we conduct an empirical analysis on the LMSYS-Chat-1M dataset [R-9], a large-scale, real-world multi-turn conversation dataset containing one million conversations between real users and 25 state-of-the-art LLMs, collected from the Vicuna demo and Chatbot Arena website between April and August 2023. As shown in Table 6, only 54.8% of user questions appear at the end of user inputs, while a substantial 45.2% are distributed in non-end positions, including 23.4% at the beginning and 21.8% in the middle.
>
> We have included the details of these discussions in Appendix A.4.1.
>
> Table 5: Existing multi-turn benchmarks
>
> | Dataset                 | Avg. # Tokens/ Turn | Avg. # Turns |
> | ----------------------- | ------------------- | ------------ |
> | Multi-IF\[R-4\]         | 19.02               | 2.99         |
> | FairMT-Bench(1K)\[R-5\] | 29.66               | 5            |
> | MT-Bench\[R-6\]         | 33.49               | 2            |
> | MultiChallenge\[R-7\]   | 41.18               | 5.06         |
> | MT-Eval\[R-8\]          | 60.63               | 6.96         |
> | StructFlowBench\[R-10\] | 93.66               | 4.15         |
> | Daily-MTD\[R-11\]       | 95.33               | 2.83         |
> | MARS-Bench\[R-12\]      | 183.34              | 33.42        |
> | Ours                    | 9321.49             | 2.61         |
>
> Table 6:  The question position distribution in LMSYS-Chat-1M dataset.
>
> | Question Position                       | Percentage      |
> | --------------------------------------- | --------------- |
> | Begin                                   | 23.4%           |
> | Middle                                  | 21.8%           |
> | End                                     | 54.8%           |
> | **Non-terminal (Front + Middle)** | **45.2%** |
>
> \[R-4\] Y. He et al., “Multi-IF: Benchmarking LLMs on Multi-Turn and Multilingual Instructions Following.” 2024.
>
> \[R-5\] Z. Fan, R. Chen, T. Hu, and Z. Liu, “FairMT-Bench: Benchmarking Fairness for Multi-turn Dialogue in Conversational LLMs.” 2025.
>
> \[R-6\] L. Zheng et al., “Judging LLM-as-a-Judge with MT-Bench and Chatbot Arena.” 2023.
>
> \[R-7\] V. Sirdeshmukh et al., “MultiChallenge: A Realistic Multi-Turn Conversation Evaluation Benchmark Challenging to Frontier LLMs.” 2025.
>
> \[R-8\] W.-C. Kwan et al., “MT-Eval: A Multi-Turn Capabilities Evaluation Benchmark for Large Language Models.” 2024.
>
> \[R-9\] L. Zheng et al., “LMSYS-Chat-1M: A Large-Scale Real-World LLM Conversation Dataset.” 2023.
>
> \[R-10\] J. Li, J. Li, Y. Wang, Y. Chang, and Y. Wu, “StructFlowBench: A Structured Flow Benchmark for Multi-turn Instruction Following.” 2025.
>
> \[R-11\] Y. Tang et al., “Learning an Efficient Multi-Turn Dialogue Evaluator from Multiple Judges.” 2025.
>
> \[R-12\] C. Yang et al., “MARS-Bench: A Multi-turn Athletic Real-world Scenario Benchmark for Dialogue Evaluation.” 2025.
>
> **W1-(3)**: Please add more details about the construction of the multi-turn data.
>
> **Reply to W1-(3):**
> We have added more details about the multi-turn data construction in Appendix A.4.4.
> First, we provide detailed pseudocode for each task type, clearly illustrating our systematic construction pipeline from LongBench to our multi-turn benchmark. Regarding reliability, the pseudocode demonstrates how we validate question-context dependencies: each question depends only on the current and previous turns' inputs, ensuring no future information leakage and maintaining logical consistency across multi-turn dialogues. Please refer to Appendix A.4.4 for more details.

---

> ### Author Response · Authors · 2025-11-25
> **Reply to W2, W3, and W4**
>
> **W2:** Please add ablation studies on the inference time of the proposed LoopServe.
>
> **Reply to W2:** We have added an ablation study to investigate the inference time of our model, which is included in Appendix A.9. In addition to the full LoopServe model, we include two variants: LoopServe-P, which is equipped only with the sparsification prefilling technique (see Section 6.1), and LoopServe-D, which utilizes only progressive decoding (see Section 6.1).
>
> As shown in the following table, with input lengths varying over {2048, 4096, 8192, 16384, 32768}, we observe that LoopServe and its variants are more efficient than the base model. However, the full LoopServe model is slightly less efficient than LoopServe-P, as it needs to select the KV cache multiple times. Nevertheless, LoopServe achieves the best effectiveness among all variants.
>
> Table 7: Ablation study of inference time (seconds).
>
> | Methods vs. End-to-end Latency | 2048           | 4096           | 8192           | 16384          | 32768          |
> | ------------------------------- | -------------- | -------------- | -------------- | -------------- | -------------- |
> | Base                            | 6.77           | 12.67          | 26.36          | 56.95          | 135.52         |
> | LoopServe                       | 2.97           | 3.37           | 4.43           | 7.58           | 15.81          |
> | LoopServe-P                     | **1.82** | **1.90** | **2.46** | **3.20** | **6.16** |
> | LoopServe-D                     | 4.55           | 5.05           | 6.10           | 9.25           | 17.26          |
>
> **W3:** (1) Please state the motivation for building multi-turn data from single-turn datasets.
> (2) It is better to include more turns instead of just 3 turns.
>
>  **Reply to W3:**  (1) We build multi-turn data from single-turn datasets because existing multi-turn datasets are typically too short per turn and do not reflect the long-context and complex dependencies seen in real-world applications (see Appendix Table 4). By systematically converting high-quality single-turn datasets into a multi-turn format, we can create benchmarks with realistic long-context inputs, diverse question positions, and explicit turn dependencies, filling an important gap for the evaluation of LLMs in practical multi-turn dialog scenarios.
>
> (2) We agree that incorporating more turns could provide a more comprehensive evaluation. Our primary choice of 3-turn dialogs was made to balance computational cost with realistic conversation lengths, as most real-world multi-turn dialogs tend to have a small number of extended turns (see Table 5).
>
> However, our data generation procedure readily supports longer dialogs, as shown in Appendix A.11. For the rebuttal, we generate a from 1 to 9-turn  dataset based on Qasper data and select strong baselines for comparison, including SnapKV, AdaKV, and Minference. The hypermeter setting is same as main experiments. As shown in the Table 8 below, even with these longer interactions, LoopServe remains effective at capturing inter-turn dependencies.
>
> Table 8: evaluation on from 1 to 9-turn multi-turn datasets.
>
> | Turns      | 1               | 2              | 3               | 4               | 5               | 6               | 7               | 8               | 9               | Average         |
> | ---------- | --------------- | -------------- | --------------- | --------------- | --------------- | --------------- | --------------- | --------------- | --------------- | --------------- |
> | SnapKV     | 16.63           | 15.18          | 14.04           | 16.23           | 16.02           | 15.87           | 16.05           | 16.68           | 15.26           | 15.77           |
> | AdaKV      | 16.53           | 15.49          | 15.42           | 18.69           | 19.13           | 17.82           | 18.77           | 18.94           | 18.52           | 17.70           |
> | Minference | 17.97           | 19.91          | 20.26           | 21.46           | 22.57           | 22.40           | 20.43           | 19.98           | 22.28           | 20.81           |
> | LoopServe  | **21.36** | **22.8** | **22.86** | **25.93** | **26.47** | **23.40** | **23.36** | **24.04** | **25.50** | **23.97** |
>
> **W4:** The readability should be improved, including the algorithm descriptions and the explanations of question positions.
>
>  **Reply to W4:**
> (1) We have added further clarification in the paper.
> (2) The "begin," "middle," and "end" positions refer to the placement of the user question within the input. We have explicitly included explanations in the main experiment table caption and provided illustrative figures in Appendix A.4.2.

---

> ### Author Response · Authors · 2025-11-25
> **Reply to W5 and Q1**
>
> **W5:** No implementation information is provided, and no information about how to use the dataset for evaluation, e.g., the split.
>
>  **Reply to W5:** We implement our method with HuggingFace Transformers as a backbone, and use monkeypatch to replace the attention forward function. We implement our method based on MInference's sparse attention functions. You can find our released code [here](https://anonymous.4open.science/r/LoopServe). This link is also included in the supplementary materials.
>
> For dataset usage, we provide user-friendly code and clear instructions in our [data repository](https://huggingface.co/datasets/TreeAILab/Multi-turn_Long-context_Benchmark_for_LLMs) to help users easily load and utilize our dataset for LLM evaluation.
>
> Regarding data splits, since our benchmark is designed exclusively for inference evaluation rather than training, we use the entire dataset for evaluation purposes. This is consistent with the standard practice in other inference-focused benchmarks, such as [LongBench](https://github.com/THUDM/LongBench)\[R-13\], where all data are included in the evaluation set.
>
> \[R-13\] Bai, Y., Lv, X., Zhang, J., Lyu, H., Tang, J., Huang, Z., Du, Z., Liu, X., Zeng, A., Hou, L., Dong, Y., Tang, J., & Li, J. (2024). LongBench: A Bilingual, Multitask Benchmark for Long Context Understanding.
>
> **Q1:**  What is the main difference compared to existing inference-time studies in terms of multi-turn scenarios for both efficiency and effectiveness?
>
> **Reply to Q1:**
> The main difference between LoopServe and existing inference-time studies lies in our focus on realistic multi-turn scenarios with long contexts and varied question positions. Previous works typically evaluate efficiency and effectiveness on benchmarks with short conversations and questions always at the end of user inputs, which allows simple position-based heuristics but fails to capture the complexity of real-world dialogues. In contrast, LoopServe is evaluated on benchmarks that include much longer contexts, diverse question placements, and explicit cross-turn dependencies, requiring adaptive methods. Our approach introduces online attention sparsification and progressive KV compression that dynamically adapt to changing dialogue patterns, ensuring robust efficiency and effectiveness where prior methods often degrade.

---

### Official Review · Reviewer_9Dbm · 2025-11-07

**Soundness:** 3
**Presentation:** 3
**Contribution:** 3
**Rating:** 6
**Confidence:** 2

**Summary:**

This paper tackles efficient LLM inference for long, multi-turn dialogues, arguing that fixed or position-based heuristics used by many acceleration methods fail under realistic, shifting attention patterns. The authors propose LoopServe, a dual-phase system: (i) online prefilling sparsification that dynamically selects high-mass “vertical” and “slash” lines in the attention matrix to retain a target fraction of total attention weight, and (ii) progressive KV compression that repeatedly reselects important input tokens using the most recently generated outputs during decoding. They also introduce an evaluation suite of 11 multi-turn long-context datasets with queries appearing at the beginning/middle/end and cross-turn dependencies. Experiments on Llama-3.1-8B and Qwen2.5-7B show LoopServe maintains or improves quality across query positions and reduces latency, with ablations and sensitivity studies supporting the contribution.

**Strengths:**

1. Well-motivated and adaptive design. The paper presents clear empirical observations that attention is input-dependent and that query position strongly affects the success of KV heuristics; LoopServe’s two phases are designed to adapt in both prefilling and decoding to these realities.

2. Thorough evaluation and new benchmark. Results across 11 datasets and multiple tasks (QA, summarization, few-shot) demonstrate consistent effectiveness and latency gains; the benchmark’s varied query positions addresses a common weakness of existing long-context tests.

**Weaknesses:**

1. Baseline coverage. Table 1 compares against SnapKV, AdaKV, StreamingLLM, A-shape, Tri-shape, and MInference, but not H2O/Heavy-Hitter Oracle or Keyformer—both relevant KV-reduction methods the paper cites elsewhere. Including these would strengthen claims of SOTA performance.

**Questions:**

1. Fairness & tuning of baselines. Authors “use suggested settings” for baselines while LoopServe has its own α, B, and n_d. Please clarify per-baseline tuning effort and whether baselines were re-tuned for the new multi-turn benchmark and non-end query positions.

---

> ### Author Response · Authors · 2025-11-25
>
> Thank you for your valuable comments. We provide our responses to each comment below, all of which are included in our revised paper. Please do not hesitate to let us know if you have any additional questions or concerns.
>
> **Reply to W1:** Thank you for the suggestion. We have conducted an additional experiment using H2O/Heavy Hitter Oracle on Llama-3.1-8B-Instruct. The results, shown below, are lower than those of our proposed LoopServe method. For more details, please refer to the main results in Table 1.
>
> | Qwen-2.5-7B-Instruct | MultifieldQA | 2WikiMQA | Musique | HotpotQA | NarrativeQA | Qasper | MultiNews | GovReport | QMSum | Trec  | SAMSUM |
> | -------------------- | ------------ | -------- | ------- | -------- | ----------- | ------ | --------- | --------- | ----- | ----- | ------ |
> | Begin                | 10.62        | 11.37    | 3.49    | 11.63    | 4.72        | 7.68   | 13.50     | 10.76     | 13.44 | 49.63 | 9.46   |
> | Middle               | 9.71         | 7.97     | 1.57    | 7.88     | 5.15        | 8.02   | 13.01     | 11.12     | 14.00 | 39.07 | 9.15   |
> | End                  | 22.64        | 29.70    | 7.68    | 25.21    | 10.64       | 14.17  | 14.53     | 8.95      | 17.93 | 58.29 | 37.37  |
>
> | Llama3.1-8B-Instruct | MultifieldQA | 2WikiMQA | Musique | HotpotQA | NarrativeQA | Qasper | MultiNews | GovReport | QMSum | Trec  | SAMSUM |
> | -------------------- | ------------ | -------- | ------- | -------- | ----------- | ------ | --------- | --------- | ----- | ----- | ------ |
> | Begin                | 23.37        | 20.77    | 11.91   | 26.49    | 6.20        | 10.91  | 18.18     | 18.58     | 16.53 | 45.48 | 17.80  |
> | Middle               | 20.95        | 16.18    | 9.72    | 26.94    | 5.55        | 13.43  | 18.02     | 18.44     | 15.23 | 41.71 | 9.88   |
> | End                  | 31.76        | 36.51    | 21.79   | 45.46    | 19.93       | 18.78  | 18.37     | 19.08     | 19.83 | 56.53 | 24.22  |
>
> We attempt to conduct our experiment on KeyFormer. However, the open-source KeyFormer code only supports the following models: cerebras/Cerebras-GPT-6.7B, mosaicml/mpt-7b, and EleutherAI/gpt-j-6B. The code implementations for these models differ significantly from those of Llama and Qwen, making it very difficult to adapt KeyFormer to these architectures. We acknowledge the contributions of KeyFormer and cite it accordingly.
>
> Alternatively, we compare our method to the latest KV compression model, SnapKV, which is published after KeyFormer. Our LoopServe method significantly outperforms SnapKV, indicating its superior performance.
>
> **Reply to Q1:** Thank you for raising this important point regarding the tuning and fairness of baseline comparisons.
>
> For each baseline, we followed the hyperparameter recommendations from the respective original papers. When ranges or options were provided, we selected the configuration that yielded the best performance on our dataset. When default settings are provided instead of ranges, we use the default settings. The settings used to conduct experiements are detailed in Appendix A.8.

---

### Author Response · Authors · 2025-12-03
**Rebuttal Summary to Area Chair**

Dear Area Chair,

Thank you very much for your and the reviewers’ careful evaluations and constructive feedback. We have thoroughly addressed all concerns and made significant improvements to the manuscript. We respectfully request your consideration of our submission for acceptance at ICLR 2026, and would like to briefly highlight our main contributions below.


- **Novelty:** We present LoopServe, a dual-phase and adaptive LLM inference acceleration system explicitly designed for long-context, multi-turn dialogues.  We propose online attention sparsification during the prefilling stage with progressive,  and output-aware progressive KV cache compression during decoding, enabling adaptive and efficient inference in complex conversational scenarios.

- **Realism and Benchmarking:** We identify and empirically validate a critical bias in existing long-context benchmarks: they overwhelmingly place user questions only at the end of user inputs and focus on short multi-turn contexts (typically fewer than 200 tokens per turn). To address this, we construct and release a new public benchmark spanning 11 datasets, designed with realistic and diverse question positions, including beginning, middle, and end of user inputs, as well as explicit cross-turn dependencies. This design is grounded in quantitative analysis of real-world conversational logs (LMSYS-Chat-1M), ensuring our benchmark more accurately reflects practical dialogue scenarios.


- **Empirical Results:** LoopServe consistently outperforms strong baselines such as SnapKV, AdaKV, and Minference in both efficiency and generation quality, particularly when queries appear in non-terminal positions, which is a common occurrence in practice. As our analysis shows that 45.2% of user queries fall into this category. Our evaluation includes comprehensive ablation studies, sensitivity analyses, and detailed breakdowns of resource usage.


In addition, we have addressed all of the reviewers’ comments and incorporated the following improvements into the revised manuscript:

- **More Baselines:** We have extended our experiments to include additional baselines, such as H2O, clarified our tuning strategies for all baselines, and documented all relevant hyperparameters.

- **Synthetic Benchmark and Realism:**  Our benchmark construction is guided by empirical analysis of real conversation logs (LMSYS-Chat-1M), which shows that nearly half of user queries are not at the end of the input. We provide detailed construction algorithms, comprehensive data statistics, and dependency checks in Appendix A.4 to ensure realism and reliability.

- **Efficiency and Overhead:** We present comprehensive comparisons of latency and memory usage across varying input and generation lengths, as well as different KV budgets. These results demonstrate that LoopServe is both practical and competitive in terms of resource usage, with clear operational breakdowns provided in Appendix A.13.

- **Parameter Sensitivity and Implementation Details:** We include detailed ablations for observation window size, reselection interval, and other parameters (Appendix A.10), as well as clarifications of our codebase and dataset usage, which are now publicly available. We demonstrate LoopServe’s effectiveness on both large and small models, under memory constraints.

In summary, LoopServe addresses a pressing gap in LLM inference for realistic and complex multi-turn dialogues, offering both methodological novelty and clear practical impact. Our open-source benchmark and reproducible code are intended to provide a robust foundation for future research in this area. All reviewer comments have been thoroughly addressed through new experiments, additional clarifications, and enhanced documentation.

We appreciate your consideration and are happy to provide further clarifications if needed.

Thank you for your time and attention.

Warm Regards,

Authors

---

### Meta-Review · Area_Chair_x5ee · 2026-01-13

**Summary:**

The reviewers broadly agree that LoopServe tackles an important and timely problem—efficient inference for long-context, multi-turn LLM dialogues—and that the core idea of adaptive, dual-phase acceleration is novel and well motivated. Strengths repeatedly noted include the adaptive design, strong empirical motivation (attention sparsity and query-position bias), and comprehensive evaluations across multiple datasets and models.

However, concerns centered on four main areas:

- Baseline coverage and fairness: Some reviewers questioned whether all relevant baselines (e.g., H2O/Heavy-Hitter Oracle, KeyFormer, improved StreamingLLM variants) were included and whether hyperparameters were fairly tuned for the new benchmark.

- Benchmark realism and construction: Skepticism was raised about the synthetic nature of the proposed multi-turn benchmark, whether it reflects real-world dialogues, and whether building multi-turn data from single-turn datasets is justified.

- Efficiency and overhead clarity: Several reviewers wanted clearer evidence that the additional online sparsification and progressive KV reselection do not negate latency gains, along with clearer breakdowns of prefilling vs. decoding costs.

- Presentation and clarity: Lower-score reviewers highlighted issues with readability, missing implementation details, unclear complexity analysis, and insufficient explanation of dataset construction and evaluation protocols.

These concerns led to mixed scores: two reviewers were marginally above the acceptance threshold, one slightly below, and one clearly negative. The rebuttal was therefore critical in determining whether the paper clears the bar.

**Reviewer Concerns:**

Largely Addressed by the Rebuttal

- Baseline coverage: The authors added results for H2O/Heavy-Hitter Oracle and clearly explained the infeasibility of running KeyFormer on the chosen architectures, while justifying comparison against SnapKV as a stronger, newer baseline.

- Fairness and tuning: Detailed clarification of hyperparameter choices and tuning strategies was provided, with explicit references to appendix sections listing configurations.

- Benchmark realism: The rebuttal directly addressed realism concerns by analyzing the LMSYS-Chat-1M dataset, showing that ~45% of user questions occur in non-terminal positions, thereby empirically justifying the benchmark design.

- Efficiency and overhead: Extensive new tables report latency and memory usage across context lengths, generation lengths, KV budgets, and ablation variants (prefilling-only, decoding-only), plus a per-stage latency breakdown demonstrating practical overhead.

I- mplementation details and reproducibility: Code release, dataset access, construction pseudocode, and evaluation protocols were clarified and linked.

- Presentation issues: Reported formatting errors, missing explanations, and unclear parameter choices were corrected, with added sensitivity analyses and clearer complexity separation.

Partially or Still Outstanding

- Validation on naturally occurring long-context dialogues: While the authors convincingly argue that existing real datasets are too short, there is still no direct end-to-end evaluation on truly organic long-context conversations.

- Scope of design-space exploration: Some higher-level concerns remain about whether alternative adaptive designs or stronger variants of baselines could further narrow the gap.

Ultra-long contexts and deployment scale: Experiments beyond ~100K tokens and broader deployment scenarios remain future work, as acknowledged by the authors.

**Reviewer Scores:**

Reviewer 9Dbm (initial score: 6, marginal accept)
Likely unchanged. Baseline coverage and tuning concerns—their main weakness—were directly addressed.

Reviewer iTYk (initial score: 2, reject)
Likely increased significantly (2 → 4). Many core complaints (efficiency alignment, benchmark motivation, ablations, implementation details, readability) were explicitly resolved, though they may still question benchmark realism.

Reviewer e2Wf (initial score: 4, marginal reject)
Likely improved to borderline accept (4 → 6). Real-world query-position analysis, overhead breakdowns, and observation-window experiments address most stated weaknesses, leaving only broader realism concerns.

Reviewer Qgxd (initial score: 6, marginal accept)
Likely stable. Most technical and presentation concerns were handled with added experiments, clarifications, and low-resource evaluations.

---

### Decision · Program_Chairs · 2026-01-26

Reject